# Adjoint Matching through the Lens of the Stochastic Maximum Principle in Optimal Control

## Abstract

Reward fine-tuning of diffusion and flow models and sampling from tilted or Boltzmann distributions can both be formulated as stochastic optimal control (SOC) problems, where learning an optimal generative dynamics corresponds to optimizing a control under SDE constraints. In this work, we revisit and generalize *Adjoint Matching*, a recently proposed SOC-based method for learning optimal controls, and place it on a rigorous footing by deriving it from the *Stochastic Maximum Principle* (SMP). We formulate a general Hamiltonian adjoint matching objective for SOC problems with control-dependent drift and diffusion and convex running costs, and show that its expected value has the same first variation as the original SOC objective. As a consequence, critical points satisfy the Hamilton–Jacobi–Bellman (HJB) stationarity conditions. In the important practical case of state- and control-independent diffusion, we recover the *lean* adjoint matching loss previously introduced in adjoint matching, which avoids second-order terms and whose critical points coincide with the optimal control under mild uniqueness assumptions. Finally, we show that adjoint matching can be precisely interpreted as a continuous-time method of successive approximations induced by the SMP, yielding a practical and implementable alternative to classical SMP-based algorithms, which are obstructed by intractable martingale terms in the stochastic setting. These results are also of independent interest to the stochastic control community, providing new implementable objectives and a viable pathway for SMP-based iterations in stochastic problems.

## 1 Introduction

Flow Matching (Lipman et al., 2023; Albergo & Vanden-Eijnden, 2023; Liu et al., 2023) and denoising diffusion models (Song & Ermon, 2019; Ho et al., 2020; Song et al., 2021; Kingma et al., 2021) are the state-of-the-art for many generative modeling applications, including text-to-image (Rombach et al., 2022; Esser et al., 2024), text-to-video (Singer et al., 2022), text-to-audio (Le et al., 2024; Vyas et al., 2023), and scientific tasks like protein structure prediction (Abramson et al., 2024). The generation process involves solving ordinary or stochastic differential equations (ODEs/SDEs) with vector fields pretrained on large-scale datasets. Oftentimes, the pretrained base model does not achieve the desired sample quality. If we have access to a reward model $r(x)$ that assesses the sample quality, a common remedy is to post-train, or *fine-tune*, the base model in order to sample from the *tilted distribution* with density:

$$p^*(x) \propto p_{\text{base}}(x) \exp(r(x)), \tag{1}$$

where $p_{\text{base}}$ denotes the density of the distribution induced by the pretrained base model.

Flow matching and diffusion models are typically pre-trained using samples from the distribution that we want to model (images, audio, video, etc.). A related problem that is central in the computational sciences is sampling from the Boltzmann distribution, defined by an unnormalized energy function $E$, i.e.

$$p^*(x) \propto \exp(-\beta E(x)), \tag{2}$$

where $\beta > 0$ denotes the inverse temperature. This fundamental task has wide applications in molecular modeling and Bayesian inference. Markov Chain Monte Carlo (MCMC, Neal (2001; 2011)) and Sequential Monte Carlo (SMC, Del Moral et al. (2006)) are classical approaches that have been studied extensively, but often suffer from slow mixing and poor scalability to high-dimensional settings. Albergo et al. (2019); Arbel et al. (2021); Gabrié et al. (2022) augment Monte Carlo methods using normalizing flows (Rezende & Mohamed, 2015; Chen et al., 2018), and building on these ideas, a more recent line of work Tzen & Raginsky (2019); Zhang & Chen (2022); Berner et al. (2023); Bruna & Han (2024); Richter & Berner (2024); Havens et al. (2025); Ren et al. (2026) consider generation algorithms based on continuous normalizing flows, i.e. samples are generated by solving ODEs or SDEs with learned vector fields.

As we review in Section 2, both reward fine-tuning and Boltzmann sampling can be cast as stochastic optimal control (SOC) problems. In the high-dimensional settings typical of these applications, classical grid-based methods for SOC are intractable, and the control is usually parameterized by a neural network and optimized via stochastic gradient methods. Domingo-Enrich et al. (2025) proposed *adjoint matching*, an SOC method that frames learning the optimal control as an $L^2$ regression problem in which the target is computed by solving an adjoint ODE backward in time. This approach was applied successfully to reward fine-tuning of a text-to-image flow matching model, yielding empirical improvements over existing methods at the time.

Despite its empirical success, adjoint matching was originally proposed heuristically within the control-affine quadratic-cost setting, and its optimality was verified a posteriori rather than derived from first principles (see Section 2.3). This heuristic construction does not reveal the underlying control-theoretic structure that explains *why* the simplification preserves optimality, nor does it extend to more general SOC formulations.

This limitation is not merely formal. Recent generative modeling work has begun to use non-Gaussian forward processes with state-dependent diffusion, including simplex diffusions based on Cox–Ingersoll–Ross or Jacobi-type processes (Richemond et al., 2022; Avdeyev et al., 2023) and multiplicative models based on geometric Brownian motion (Shetty et al., 2025; Kim et al., 2025). More recently, Tang et al. (2026) developed unified denoising objectives for several such processes. Reward fine-tuning of models with these dynamics therefore calls for an adjoint-matching theory that applies beyond the time-dependent-diffusion setting of the original formulation.

In this paper, we revisit and generalize adjoint matching, and establish its connection to stochastic optimal control through the *stochastic maximum principle* (SMP). Beyond the generative modeling context, these results are also of independent interest to the stochastic control community, as they yield new implementable algorithms for SOC and a practical resolution to a long-standing obstacle in SMP-based methods. More broadly, methods for stochastic optimal control are increasingly central to machine learning: in addition to reward fine-tuning of diffusion and flow models, adjoint-matching-based algorithms have recently been used for sampling from unnormalized densities (Havens et al., 2025; Liu et al., 2025) and for reinforcement learning in robotics (Li & Levine, 2026). Our results connect this growing family of machine-learning algorithms with the classical stochastic control literature, and are intended to facilitate exchange between the two communities. Our main contributions are as follows:

- **A general Hamiltonian BAM formulation for SOC (Theorem 1).** We formulate a *general Hamiltonian basic adjoint matching* (BAM) objective for SOC problems with general control-dependent drift and diffusion and convex running costs. This formulation relies on unbiased estimators of the cost-to-go gradient and Hessian via first- and second-order adjoint equations, and substantially extends the setting of Domingo-Enrich et al. (2025), which arises as a special case. We show that the expected BAM objective has the same first variation as the original SOC objective, and that its critical points satisfy the stationarity conditions associated with the stochastic maximum principle (SMP) and Hamilton–Jacobi–Bellman (HJB) formulations.

- **A lean adjoint matching loss for time-dependent diffusion (Theorem 2).** In the practically important case where the diffusion depends only on time, the BAM framework reduces to a *lean adjoint matching* (AM) loss that avoids second-order terms and recovers the original adjoint matching objective of Domingo-Enrich et al. (2025) in the control-affine quadratic-cost setting as a special case.

Under standard uniqueness assumptions, the critical points of this lean AM loss recover the optimal control.

- **A precise SMP interpretation of adjoint matching (Theorem 5).** The preceding results characterize the critical points of the lean AM loss but do not address whether gradient-based optimization of that loss implements a principled improvement procedure. We further show that the gradient flow induced by the expected lean AM loss coincides with a continuous-time method of successive approximations (MSA) derived from the stochastic maximum principle (SMP) for optimal stochastic control, while bypassing the intractable martingale term that obstructs classical SMP-based schemes. For parameterized controls such as neural networks, this gives gradient-based training of the lean AM loss a concrete control-theoretic interpretation, rather than treating it as a heuristic regression objective.

**Roadmap.** Section 2 reviews the SOC formulation, the original adjoint matching objective, and the stochastic maximum principle (SMP). Section 3 introduces the general adjoint matching objective from a Hamiltonian perspective (Theorem 1), then specializes to purely time-dependent diffusion to obtain a lean first-order objective whose critical points recover the optimal control (Theorem 2). Section 4 shows how the lean adjoint matching of this paper is precisely the continuous-time MSA induced by the SMP (Theorem 5). Proofs and technical extensions are deferred to the appendix.

## 2 Background

We begin by reviewing the stochastic optimal control (SOC) formulation that underlies both reward fine-tuning and Boltzmann sampling, then recall the adjoint matching objective proposed by Domingo-Enrich et al. (2025) and the maximum-principle background used later in the paper.

### 2.1 Stochastic Optimal Control Formulation

We start with the following generic SOC problem. Let $f : \mathbb{R}^d \times \mathbb{R}^k \times [0, T] \to \mathbb{R} \cup \{+\infty\}$ be proper, lower semicontinuous, and convex in its second argument, let $b : \mathbb{R}^d \times \mathbb{R}^k \times [0, T] \to \mathbb{R}^d$ and $\sigma : \mathbb{R}^d \times \mathbb{R}^k \times [0, T] \to \mathbb{R}^{d \times m}$ be $C^1$ in $x$, and $B_t$ denote $m$-dimensional Brownian motion. We aim to solve:

$$\min_{u \in \mathcal{U}} \ \mathbb{E}\left[ \int_0^T f(X_t^u, u(X_t^u, t), t) \, dt + g(X_T^u) \right], \tag{3}$$

$$\text{s.t.} \quad dX_t^u = b(X_t^u, u(X_t^u, t), t) \, dt + \sigma(X_t^u, u(X_t^u, t), t) \, dB_t, \qquad X_0^u \sim P_0. \tag{4}$$

Here $P_0$ is the initial distribution, $f$ is the running cost, $g$ is the terminal cost, $b$ is the drift, $\sigma$ is the diffusion coefficient, and $\mathcal{U}$ is the space of Markov control functions $u : \mathbb{R}^d \times [0, T] \to \mathbb{R}^k$.

### 2.2 Equivalence to Reward Fine-Tuning and Sampling

Reward fine-tuning of diffusion and flow models (problem 1) and sampling with continuous normalizing flows (problem 2) are closely connected: if we let $p_{\text{base}}$ be a Gaussian, given an energy $E$ we can construct a reward function $r(x) = -\beta E(x) - \log p_{\text{base}}(x)$ such that problem 2 reduces to problem 1. Moreover, both problems are equivalent to SOC problems. Generically, the control-affine quadratic-cost SOC problem reads

$$\min_{u \in \mathcal{U}} \mathbb{E}\left[ \int_0^T \tfrac{1}{2} \|u(X_t^u, t)\|^2 + f(X_t^u, t) \, dt + g(X_T^u) \right], \tag{5}$$

$$\text{s.t.} \quad dX_t^u = (b(X_t^u, t) + \sigma(X_t^u, t) u(X_t^u, t)) \, dt + \sigma(X_t^u, t) dB_t, \quad X_0^u \sim P_0. \tag{6}$$

Here the symbols are as in equation (3), with $b$ now denoting the base (uncontrolled) drift and $f$ the state-dependent running cost (as opposed to the joint running cost $f(x, u, t)$ above). According to the path integral formulation of SOC (Kappen (2005), see (Domingo-Enrich et al., 2024, Eq. 8, App. B) for a self-contained proof), the Radon-Nikodym derivative between the probability measures of the optimally controlled process $\mathbb{P}^\star$ and the uncontrolled process $\mathbb{P}^{\text{base}}$ ($u \equiv 0$) is equal to

$$\frac{d\mathbb{P}^\star}{d\mathbb{P}^{\text{base}}}(X) = \exp\left( -\int_0^T f(X_t, t) \, dt - g(X_T) + V(X_0, 0) \right), \tag{7}$$

with $X$ denoting the uncontrolled process and $V$ denoting the *value function* or *cost-to-go*, defined by

$$V(x, t) = -\log \mathbb{E}[\exp(-\int_t^T f(X_s, s)\mathrm{d}s - g(X_T)) \mid X_t = x].$$

Domingo-Enrich et al. (2025, Sec. 4.2, 4.3) show that when $f \equiv 0$ and the base process satisfies the *memorylessness* condition[1], equation (7) implies that

$$p^*(x) \propto p^{\mathrm{base}}(x) \exp(-g(x)), \tag{8}$$

where $p^*$ and $p^{\mathrm{base}}$ are the marginals of $\mathbb{P}^\star$ and $\mathbb{P}^{\mathrm{base}}$ at terminal time $t = T$, respectively. A comparison between equation (8) and equations (1)(2) shows that we can recast reward fine-tuning and sampling into the SOC problem equation (5) by setting (i) $f \equiv 0$, (ii) the distribution $P_0$ and diffusion coefficient $\sigma$ such that $X$ is memoryless, and (iii) the terminal cost as $g(x) = -r(x)$, $g(x) = \beta E(x) + \log p_{\mathrm{base}}(x)$, respectively.

The SOC viewpoint is one of several frameworks for reward fine-tuning of diffusion models. An alternative line of work treats the generation process as a discrete-time Markov decision process and applies reinforcement learning techniques such as policy gradient algorithms (Fan et al., 2023; Black et al., 2024). Other methods directly backpropagate the reward gradient through the sampling chain (Clark et al., 2024; Prabhudesai et al., 2023). By contrast, adjoint matching and the methods studied in this paper operate in the continuous-time SOC framework equation (5) (Uehara et al., 2024; Tang & Zhou, 2024; Han et al., 2025; Gao et al., 2026), leveraging adjoint equations to compute the optimal control without backpropagating through the full sampling procedure.

### 2.3 The Original Adjoint Matching Objective

Domingo-Enrich et al. (2025) proposed *adjoint matching* for the control-affine quadratic-cost SOC problem equation (5) with purely time-dependent diffusion $\sigma(x, u, t) = \sigma(t)$. In this setting, they define the *lean adjoint* $\tilde{a}(t; X) \in \mathbb{R}^d$ as the solution of the backward ODE

$$\frac{\mathrm{d}\tilde{a}}{\mathrm{d}t}(t; X) = -\nabla_x b(X_t, t)^\top \tilde{a}(t; X) - \nabla_x f(X_t, t), \tag{9}$$

$$\tilde{a}(T; X) = \nabla_x g(X_T), \tag{10}$$

and the adjoint matching loss is

$$\mathcal{L}_{\mathrm{AM}}(u; X^{\bar{u}}) = \int_0^T \frac{1}{2}\|u(X_t^{\bar{u}}, t) + \sigma(t)^\top \tilde{a}(t; X^{\bar{u}})\|^2 \, \mathrm{d}t, \qquad \bar{u} = \mathrm{stopgrad}(u). \tag{11}$$

In words, $\mathcal{L}_{\mathrm{AM}}$ is an $L^2$ regression loss that trains the control $u$ to match the target $-\sigma(t)^\top \tilde{a}(t; X^{\bar{u}})$ along trajectories simulated under the current (stop-gradiented) control $\bar{u}$.

This objective was motivated by adjoint sensitivity analysis but derived heuristically: the lean adjoint $\tilde{a}$ is a simplified version of the full adjoint process, obtained by omitting terms whose expectation vanishes at the optimal control $u^*$. Domingo-Enrich et al. (2025) proved that $u^*$ is the unique critical point of the resulting loss, but the derivation relies on verifying this property a posteriori rather than constructing the objective from first principles. In the following sections, we provide such a construction via the Hamiltonian formulation of SOC and generalize the objective to settings with arbitrary control-dependent drift and running costs (Section 3). We further show that the resulting lean adjoint matching admits an interpretation as a continuous-time MSA induced by the SMP (Section 4).

### 2.4 Maximum Principles

We next recall the control-theoretic background needed to interpret adjoint matching as a method of successive approximations (MSA): the deterministic Pontryagin maximum principle, the resulting successive-approximation algorithm, and the stochastic maximum principle. The key point for what follows is that the stochastic adjoint equation contains an unknown martingale term, which makes the direct stochastic analogue of this algorithm difficult.

---

[1]The uncontrolled process $X$ from $\mathrm{d}X_t = b(X_t, t)\mathrm{d}t + \sigma(X_t, t)\mathrm{d}B_t$ is memoryless when the random variables $X_0$ and $X_T$ are independent. See (Domingo-Enrich et al., 2025, Sec. 4.3) for sufficient conditions arising from generative modeling that guarantee this.

**Pontryagin maximum principle and method of successive approximations.** The Pontryagin maximum principle (PMP) provides necessary optimality conditions for deterministic control problems of the form

$$\min_{u \in \mathcal{U}} \ \mathbb{E} \left[ \int_0^T f(X_t^u, u(X_t^u, t), t) \, \mathrm{d}t + g(X_T^u) \right], \tag{12}$$

$$\text{s.t.} \quad \mathrm{d}X_t^u = b(X_t^u, u(X_t^u, t), t) \, \mathrm{d}t, \qquad X_0^u \sim P_0. \tag{13}$$

Introducing the *simplified Hamiltonian*

$$\tilde{H}(x, t; u, p) = f(x, u, t) + \langle b(x, u, t), p \rangle, \tag{14}$$

the PMP takes the form

$$\begin{cases} \mathrm{d}X_t^* = b(X_t^*, u_t^*, t) \, \mathrm{d}t, & (15) \\ \mathrm{d}p_t^* = -\nabla_x \tilde{H}(X_t^*, t; u_t^*, p_t^*) \, \mathrm{d}t, & (16) \\ X_0^* \sim P_0, \qquad p_T^* = \nabla_x g(X_T^*), & (17) \\ \tilde{H}(X_t^*, t; u_t^*, p_t^*) = \min_{u \in \mathbb{R}^k} \tilde{H}(X_t^*, t; u, p_t^*). & (18) \end{cases}$$

A crucial feature of the PMP is that all quantities are fully specified: the forward state ODE and the backward adjoint ODE have known coefficients, and the Hamiltonian minimization is a pointwise optimization. This gives rise to the method of successive approximations (MSA, Chernousko & Lyubushin (1982), (Li et al., 2018, Algorithm 1)), an iterative procedure repeating the following three steps:

1. Solve forwardly $\mathrm{d}X_t^k = b(X_t^k, u_t^k, t) \, \mathrm{d}t, \quad X_0^k \sim P_0$.

2. Solve backwardly $\mathrm{d}p_t^k = -\nabla_x \tilde{H}(X_t^k, t; u_t^k, p_t^k) \, \mathrm{d}t, \quad p_T^k = \nabla_x g(X_T^k)$.

3. Solve the Hamiltonian optimization condition:

$$u_t^{k+1} \in \arg\min_{u \in \mathbb{R}^k} \{\tilde{H}(X_t^k, t; u, p_t^k)\}. \tag{19}$$

Each step of MSA is implementable, and the algorithm has well-studied convergence properties in the deterministic setting (Li et al., 2018).

**Stochastic maximum principle.** The SMP generalizes the PMP to stochastic control problems. In what follows, we restrict attention to the first-order version of SMP when the diffusion coefficient does not depend on the control, $\sigma(x, u, t) := \sigma(x, t)$, which captures the regime needed for the MSA connection in Section 4. The full second-order formulation of SMP for general control-dependent diffusion is summarized in Appendix D.2 for completeness.

To state the SMP, we introduce another Hamiltonian that couples the backward adjoint $p$ with the martingale integrand $q$:

$$\mathcal{H}(x, t; u, p, q) = f(x, u, t) + \langle p, b(x, u, t) \rangle + \mathrm{tr}(\sigma(x, t)^\top q), \tag{20}$$

$$\text{where } (x, t, u, p, q) \in \mathbb{R}^d \times [0, T] \times \mathbb{R}^k \times \mathbb{R}^d \times \mathbb{R}^{d \times m}. \tag{21}$$

For the SOC problem (5), SMP states the following necessary condition for optimality. For consistency with the Hamiltonian formulation developed in Section 3, we adopt the *minimization* convention (i.e. the Hamiltonian is minimized over $u$): given an optimal control $u_t^* := u^*(X_t^*, t)$ and its corresponding state process $X_t^*$, there exist adapted processes $p_t^* \in \mathbb{R}^d$ and $q_t^* \in \mathbb{R}^{d \times m}$ such that

$$\begin{cases} \mathrm{d}X_t^* = b(X_t^*, u_t^*, t) \, \mathrm{d}t + \sigma(X_t^*, t) \, \mathrm{d}B_t, & (22) \\ \mathrm{d}p_t^* = -\nabla_x \mathcal{H}(X_t^*, t; u_t^*, p_t^*, q_t^*) \, \mathrm{d}t + q_t^* \, \mathrm{d}B_t, & (23) \\ X_0^* \sim P_0, \qquad p_T^* = \nabla_x g(X_T^*), & (24) \\ \mathcal{H}(X_t^*, t; u_t^*, p_t^*, q_t^*) = \min_{u \in \mathbb{R}^k} \mathcal{H}(X_t^*, t; u, p_t^*, q_t^*). & (25) \end{cases}$$

Here 'adapted' means $p_t$ and $q_t$ depend only on the Brownian history up to time $t$; more precisely, they are $\mathcal{F}_t$-measurable, where $(\mathcal{F}_t)$ is the filtration generated by $B$. Note that since $\sigma(X_t^*, t)$ does not depend on $u$, the trace term $\text{tr}(\sigma(X_t^*, t)^\top q_t^*)$ drops out of the minimization and equation (25) is equivalently $\tilde{H}(X_t^*, t; u_t^*, p_t^*) = \min_u \tilde{H}(X_t^*, t; u, p_t^*)$, recovering the same pointwise condition (18) as in the PMP.

Comparing the SMP with the PMP reveals a fundamental structural difference. In the PMP, the backward adjoint equation equation (16) is an ODE with known coefficients. In the SMP, the backward adjoint equation equation (23) contains an additional stochastic term $q_t^* \, dB_t$, where the diffusion coefficient $q_t^*$ is not prescribed a priori but must be solved jointly with $p^*$ so that the pair is adapted to the underlying filtration. Equations of this type are *backward stochastic differential equations* (BSDEs) Pardoux & Peng (2005), in which both the backward state and its martingale integrand are unknowns determined implicitly by the terminal condition and the dynamics. As we will see in Section 3, this stands in contrast to the basic adjoint matching framework introduced there, whose adjoint SDEs have explicit diffusion coefficients and can therefore be solved pathwise.

In the context of SMP, as shown in Appendix D.1, the process (23) admits a concrete interpretation:

$$p_t^* = \nabla_x V(X_t^*, t), \qquad q_t^* = \nabla_x^2 V(X_t^*, t)\sigma(X_t^*, t), \tag{26}$$

where $V(x, t) := J(u^*; x, t)$ is the value function. In particular, $q_t^*$ encodes second-order value function information, which is precisely the quantity that the lean adjoint matching loss avoids computing. This unknown martingale integrand is the main feature distinguishing the stochastic adjoint equation from its deterministic counterpart.

## 3 The General Adjoint Matching Loss Functions

This section derives the *adjoint matching* objectives studied in this paper from a Hamiltonian perspective. The starting point is the Hamiltonian formulation of stochastic optimal control: at an optimal control, the Hamilton–Jacobi–Bellman (HJB) stationarity conditions reduce to a pointwise minimization of the Hamiltonian, which involves the cost-to-go derivatives $\nabla_x J$ and $\nabla_x^2 J$. Adjoint matching turns this stationarity condition into a learning objective by replacing $\nabla_x J$ and $\nabla_x^2 J$ with pathwise adjoint estimators computed along simulated trajectories. Table 1 summarizes the Hamiltonians and adjoint processes used throughout.

We first treat the general SOC problem equation (3) where both drift and diffusion may depend on the control. This yields a *basic adjoint matching* (BAM) loss involving first- and second-order adjoint equations. We then specialize to the practically important setting $\sigma(x, u, t) = \sigma(t)$ (as in diffusion models), where the diffusion term does not affect the Hamiltonian minimizer; this leads to the *lean* adjoint matching (AM) loss that avoids second-order quantities, recovering the objective stated in Section 2.3 as a special case. Full regularity and well-posedness conditions are collected in Appendix B; whenever we identify a critical point with the optimal control, we additionally invoke the verification hypotheses in Lemma 10.

**Assumptions and scope.** The results below are stated under standard sufficient smoothness and integrability assumptions (Yong & Zhou, 2012): $b, \sigma, f$ are $C^1$ in $(x, u)$, $g$ is $C^1$ in $x$, the relevant derivatives have at most polynomial growth, and the controlled SDE is well posed with the moments needed to justify differentiation under the expectation. The second-order adjoint requires the corresponding $C^2$ regularity in $x$, and the verification step that identifies stationary points with the optimal control additionally assumes a classical $C^{1,2}$ cost-to-go and the convexity/uniqueness conditions stated in Lemma 10. These are sufficient conditions, not minimal ones; relaxing them, or deriving analogous guarantees after restricting the Markov controls to particular finite-dimensional model classes, is beyond the scope of this work.

Our construction starts with the full adjoint processes that provide (unbiased) access to the first-order and second-order derivatives of the cost-to-go under a fixed Markov control policy with respect to the current state. Concretely, for a fixed control $\bar{u}$, define the cost-to-go

$$J(\bar{u}; x, t) := \mathbb{E}\left[\int_t^T f(X_s^{\bar{u}}, \bar{u}(X_s^{\bar{u}}, s), s) \, ds + g(X_T^{\bar{u}}) \, \Big| \, X_t^{\bar{u}} = x\right]. \tag{27}$$

Table 1: Key Hamiltonians and adjoint processes used in this paper.

| Symbol | Description | Reference |
|---|---|---|
| HAMILTONIANS | | |
| $H(x,t;u,p,M)$ | Full HJB Hamiltonian (general $\sigma$) | Eq. (34) |
| $\tilde{H}(x,t;u,p)$ | Simplified Hamiltonian ($\sigma=\sigma(t)$; no second-order term) | Eq. (14) |
| $\mathcal{H}(x,t;u,p,q)$ | SMP Hamiltonian (includes martingale integrand $q$) | Eq. (20) |
| ADJOINT PROCESSES | | |
| $a_t$ | First-order adjoint SDE; estimates $\nabla_x J$ | Eq. (30) |
| $\tilde{a}_t$ | Lean adjoint ODE ($\sigma=\sigma(t)$); estimates $\nabla_x J$ | Eq. (40) |
| $A_t$ | Second-order adjoint SDE; estimates $\nabla_x^2 J$ | Eq. (32) |
| $(p_t^*, q_t^*)$ | SMP adjoint BSDE pair; $q_t^*$ is unknown integrand | Eq. (23) |

The adjoint processes $a_t$ and $A_t$ defined below will provide pathwise estimates for the gradient and Hessian of $J(\bar{u}; \cdot, \cdot)$ along the controlled dynamics (for a primer on first-order adjoint sensitivity for SDEs, see Leburu et al. (2026); our constructions go further by incorporating second-order adjoint processes and general control-dependent diffusion):

$$\mathbb{E}\big[a(t; X^{\bar{u}}) \mid X_t^{\bar{u}} = x\big] = \nabla_x J(\bar{u}; x, t), \tag{28}$$

$$\mathbb{E}\big[A(t; X^{\bar{u}}) \mid X_t^{\bar{u}} = x\big] = \nabla_x^2 J(\bar{u}; x, t). \tag{29}$$

To put it formally with a convenient notation, we denote by $\sigma_{\cdot k}(x, u, t) \in \mathbb{R}^d$ the $k$-th column of $\sigma(x, u, t)$ for $k \in \{1, \ldots, m\}$, and set the total derivative

$$G_k^u(x, t) := \nabla_y \sigma_{\cdot k}(y, u(y, t), t)|_{y=x} \in \mathbb{R}^{d \times d}.$$

Let $a_t := a(t; X) \in \mathbb{R}^d$ solve the first-order adjoint SDE backwardly in time:

$$\mathrm{d}a_t = \Big( -\nabla_x b(x, u(x,t), t)\big|_{x=X_t}^\top a_t - \nabla_x f(x, u(x,t), t)\big|_{x=X_t} $$
$$+ \sum_{k=1}^m G_k^u(X_t, t)^\top G_k^u(X_t, t)^\top a_t \Big) \mathrm{d}t - \sum_{k=1}^m G_k^u(X_t, t)^\top a_t \, \mathrm{d}B_t, \tag{30}$$

$$a_T = \nabla_x g(X_T). \tag{31}$$

and let $A_t := A(t; X) \in \mathbb{R}^{d \times d}$ solve the second-order adjoint SDE backwardly in time:

$$\mathrm{d}A_t = -\Big( \nabla_x b(x, u(x,t), t)\big|_{x=X_t}^\top A_t + A_t \nabla_x b(x, u(x,t), t)\big|_{x=X_t} + \sum_{k=1}^m G_k^u(X_t, t)^\top A_t G_k^u(X_t, t) $$
$$+ \nabla_x^2 f(x, u(x,t), t)\big|_{x=X_t} + \sum_{i=1}^d a_t^{(i)} \nabla_x^2 b_i(x, u(x,t), t)\big|_{x=X_t} $$
$$+ \tfrac{1}{2} \sum_{k=1}^m \sum_{i=1}^d a_t^{(i)} \nabla_x^2 \big(\sigma_{\cdot k}^{(i)}(x, u(x,t), t)\big)\big|_{x=X_t} G_k^u(X_t, t) \Big) \mathrm{d}t + \sum_{k=1}^m U_{t,k} \, \mathrm{d}B_t^k, \tag{32}$$

$$A_T = \nabla_x^2 g(X_T),$$

where $B_t^k$ denotes the $k$-th component and the diffusion matrices $U_{t,k} \in \mathbb{R}^{d \times d}$ are given explicitly by

$$U_{t,k} = -\Big( A_t \, G_k^u(X_t, t) + G_k^u(X_t, t)^\top A_t + \sum_{i=1}^d a_t^{(i)} \nabla_x^2 \big(\sigma_{ik}(x, u(x,t), t)\big)|_{x=X_t} \Big). \tag{33}$$

We note that although equation (30) and equation (32) are posed backward in time, every diffusion coefficient ($G_k^\top a_t$ and $U_{t,k}$, respectively) is given as an explicit function of the current state. These equations can therefore be solved pathwise via time-reversal. This is in contrast to the *backward stochastic differential equation (BSDE)* for the SMP adjoint introduced in Section 2.4, where an additional unknown martingale integrand $q_t$ must be determined jointly with the backward state.

With the above adjoint processes in hand, the following general basic matching loss function provides a reformulation of the original SOC problem. Its proof can be found in Appendix B.

**Theorem 1** (General Hamiltonian basic adjoint matching). *Consider the SOC problem (3)–(4). Define the Hamiltonian*

$$H(x,t;u,p,M) := f(x,u,t) + \langle b(x,u,t), p \rangle + \tfrac{1}{2}\operatorname{Tr}\big(\sigma(x,u,t)\sigma(x,u,t)^\top M\big), \tag{34}$$

$$\text{where } (x,t,u,p,M) \in \mathbb{R}^d \times [0,T] \times \mathbb{R}^k \times \mathbb{R}^d \times \mathbb{R}^{d \times d}, \tag{35}$$

*and consider the basic adjoint matching loss*

$$\mathcal{L}_{\mathrm{BAM}}(u; X^{\bar{u}}) := \int_0^T H\big(X_t^{\bar{u}}, t; u(X_t^{\bar{u}}, t), a(t; X^{\bar{u}}), A(t; X^{\bar{u}})\big)\, dt, \qquad \bar{u} = \mathrm{stopgrad}(u). \tag{36}$$

*Alternatively, $a(t; X^{\bar{u}})$ and $A(t; X^{\bar{u}})$ can be replaced by any unbiased estimates of $\nabla_x J(\bar{u}; X_t^{\bar{u}}, t)$ and $\nabla_x^2 J(\bar{u}; X_t^{\bar{u}}, t)$ (cf. (28) and (29)).*

*Then, the functional $u \mapsto \mathbb{E}[\mathcal{L}_{\mathrm{BAM}}(u; X^{\bar{u}})]$ has the same first variation as the original control objective (3). Moreover, any critical point[2] $\hat{u}$ of $u \mapsto \mathbb{E}[\mathcal{L}_{\mathrm{BAM}}(u; X^{\bar{u}})]$ satisfies the pointwise stationarity condition*

$$\nabla_u H\Big(x, t; \hat{u}(x,t), \nabla_x J(\hat{u}; x, t), \nabla_x^2 J(\hat{u}; x, t)\Big) = 0. \tag{37}$$

*If, moreover, $u \mapsto H(x, t; u, p, M)$ is convex, then stationarity is equivalent to pointwise minimization:*

$$\hat{u}(x,t) \in \arg\min_{u \in \mathbb{R}^k} H\Big(x, t; u, \nabla_x J(\hat{u}; x, t), \nabla_x^2 J(\hat{u}; x, t)\Big). \tag{38}$$

*In this case, $J(\hat{u}; \cdot, \cdot)$ satisfies the HJB equation*

$$\partial_t J + \inf_{u \in \mathbb{R}^k} H\Big(x, t; u, \nabla_x J, \nabla_x^2 J\Big) = 0, \qquad J(\cdot, T) = g. \tag{39}$$

*In particular, if the HJB solution is unique, then $\hat{u} = u^\star$ is the optimal control.*

The BAM objective (36) in Theorem 1 provides a variationally correct Hamiltonian formulation for general SOC, but its implementation requires integrating the $d \times d$ matrix SDE equation (32) for the second-order adjoint $A_t$. In the practically important setting $\sigma(x, u, t) = \sigma(t)$, standard in diffusion models, two key simplifications occur: (i) $G_k^u \equiv 0$, so the first-order adjoint SDE equation (30) reduces to an ODE; and (ii) the trace term $\frac{1}{2}\operatorname{Tr}(\sigma(t)\sigma(t)^\top A_t)$ in the Hamiltonian equation (34) no longer depends on $u$, so $A_t$ drops out of the stationarity condition, eliminating the need for the matrix SDE entirely. While this specializes the diffusion, the drift and running cost remain fully general, extending beyond the control-affine quadratic-cost setting of Domingo-Enrich et al. (2025); this yields a lean AM loss whose critical points, under a standard HJB uniqueness assumption, recover the optimal control.

**Theorem 2** (General Hamiltonian (lean) adjoint matching). *Consider the SOC problem and notation from Theorem 1, in the particular case that the diffusion coefficient depends only on time: $\sigma(x, u, t) := \sigma(t)$. Let $\nabla_1 f(x, u(x,t), t)$ and $\nabla_1 b(x, u(x,t), t)$ denote the partial derivatives with respect to the first argument $x$, as opposed to $\nabla_x f(y, u(y,t), t)\big|_{x=y}^\top$ and $\nabla_x b(y, u(y,t), t)\big|_{x=y}^\top$, which are the total derivatives. Let $\tilde{a}(t; X)$ solve the lean adjoint ODE:*

$$\tfrac{d\tilde{a}}{dt}(t; X) = -\nabla_1 b(X_t, u(X_t, t), t)^\top \tilde{a}(t; X) - \nabla_1 f(X_t, u(X_t, t), t), \tag{40}$$

$$\tilde{a}(T; X) = \nabla_x g(X_T). \tag{41}$$

*Recall the simplified Hamiltonian*

$$\tilde{H}(x, t; u, p) = f(x, u, t) + \langle b(x, u, t), p \rangle, \tag{42}$$

*from equation (14), and the (lean) adjoint matching loss*

$$\mathcal{L}_{\mathrm{AM}}(u; X^{\bar{u}}) := \int_0^T \tilde{H}\big(X_t^{\bar{u}}, t; u(X_t^{\bar{u}}, t), \tilde{a}(t; X^{\bar{u}})\big)\, dt, \qquad \bar{u} = \mathrm{stopgrad}(u). \tag{43}$$

---

[2] A critical point is a point with zero first-order variation.

*Any critical point $\hat{u}$ of $u \mapsto \mathbb{E}[\mathcal{L}_{\mathrm{AM}}(u; X^{\bar{u}})]$ satisfies the pointwise stationarity condition*

$$\nabla_u \tilde{H}\Big(x, t; \hat{u}(x,t), \nabla_x J(\hat{u}; x, t)\Big) = 0. \tag{44}$$

*If, moreover, $u \mapsto \tilde{H}(x, t; u, p)$ is convex, then stationarity is equivalent to pointwise minimization:*

$$\hat{u}(x,t) \in \arg\min_{u \in \mathbb{R}^k} \tilde{H}\Big(x, t; u, \nabla_x J(\hat{u}; x, t)\Big). \tag{45}$$

*In particular, if the HJB solution is unique, then $\hat{u} = u^\star$ is the optimal control.*

The proof is given in Appendix C. We now verify that the original adjoint matching objective of Domingo-Enrich et al. (2025) is recovered as a special case.

**Remark 3** (Recovering the original adjoint matching loss)**.** *The control-affine quadratic-cost SOC problem equation (5) studied by Domingo-Enrich et al. (2025) (cf. Section 2.3) is a special case of Theorem 2. Using the notation of Section 2.2, the drift, running cost, and diffusion in the general formulation equation (3) specialize to*

$$b(x, u, t) = b(x, t) + \sigma(t)u, \qquad f(x, u, t) = f(x, t) + \tfrac{1}{2}\|u\|^2, \qquad \sigma(x, u, t) = \sigma(t), \tag{46}$$

*where $b(x, t)$, $f(x, t)$, and $\sigma(t)$ on the right-hand side are the base drift, state cost, and diffusion coefficient from equation (5). The simplified Hamiltonian equation (14) then becomes*

$$\tilde{H}(x, t; u, p) = f(x, t) + \tfrac{1}{2}\|u\|^2 + \langle b(x, t) + \sigma(t)u, \, p \rangle, \tag{47}$$

*and the lean adjoint ODE equation (40) reduces to equation (9). Substituting into the lean adjoint matching loss equation (43) and dropping terms independent of u yields*

$$\mathcal{L}_{\mathrm{AM}}(u; X^{\bar{u}}) = \int_0^T \tfrac{1}{2}\|u(X_t^{\bar{u}}, t) + \sigma(t)^\top \tilde{a}(t; X^{\bar{u}})\|^2 \, \mathrm{d}t, \qquad \bar{u} = \mathrm{stopgrad}(u), \tag{48}$$

*which recovers exactly the adjoint matching objective in (Domingo-Enrich et al., 2025, Equations (37)–(39)) as stated in Section 2.3.*

**Remark 4** (Beyond the control-affine quadratic-cost case)**.** *Theorem 2 does more than recover the original adjoint matching objective: it extends the lean AM construction to general control dependence in the drift and general convex running costs. Specifically, the drift may be any $C^1$ function $b(x, u, t)$ of the control, and the running cost $f(x, u, t)$ may be any convex function of u, while the resulting lean AM loss keeps the same Hamiltonian-regression structure as the original algorithm: it is still an integral of the simplified Hamiltonian $\tilde{H}$ along simulated trajectories, with only $\tilde{H}$ changing. This covers settings with nonlinear control effects or non-quadratic control penalties, such as thrust-direction dynamics in robotics, non-quadratic transaction costs in portfolio optimization, or exponential control dependence in reaction-rate models, provided the diffusion is control-independent. This makes Theorem 2 directly implementable in settings where the original algorithm does not apply. The algorithmic price is that the pointwise Hamiltonian minimization need not have the closed form available in the control-affine quadratic-cost case; in general, one should expect to optimize the corresponding Hamiltonian term numerically, for example by gradient-based methods.*

## 4  Connecting Adjoint Matching to the Stochastic Maximum Principle

Having established the adjoint matching losses as principled objectives in Section 3 (building on the SMP review in Section 2.4), we now connect the two. Recall that the results of Section 3 are variational: they show that the lean AM loss has the correct critical points, so that any stationary control is optimal, but do not say whether the *optimization dynamics* used to reach those critical points are themselves principled. We now formulate the method of successive approximations (MSA) induced by the stochastic maximum principle (SMP) in the $\sigma = \sigma(t)$ setting, and then show that the gradient flow of the lean AM loss coincides with a continuous-time version of this procedure.

Consider the SMP in the case where $\sigma(x, u, t) = \sigma(t)$, which simplifies to

$$\begin{cases} \mathrm{d}X_t^* = b(X_t^*, u_t^*, t)\mathrm{d}t + \sigma(t)\mathrm{d}B_t, & (49) \\ dp_t^* = -\big(\nabla_1 f(X_t^*, u_t^*, t) + \nabla_1 b(X_t^*, u_t^*, t)^\top p_t^*\big)\,\mathrm{d}t + q_t^*\,\mathrm{d}B_t, & (50) \\ X_0^* \sim P_0, \qquad p_T^* = \nabla_x g(X_T^*), & (51) \\ 0 = \nabla_2 f(X_t^*, u_t^*, t) + \nabla_2 b(X_t^*, u_t^*, t)^\top p_t^*. & (52) \end{cases}$$

The resulting MSA reads:

1. Solve forwardly $\mathrm{d}X_t^k = b(X_t^k, u_t^k, t)\mathrm{d}t + \sigma(t)\mathrm{d}B_t, \quad X_0^k \sim P_0$.

2. Solve backwardly $dp_t^k = -\big(\nabla_1 f(X_t^k, u_t^k, t) + \nabla_1 b(X_t^k, u_t^k, t)^\top p_t^k\big)\,\mathrm{d}t + q_t^k\,\mathrm{d}B_t, \quad p_T^k = \nabla_x g(X_T^k)$.

3. Solve the Hamiltonian optimization condition:

$$u_t^{k+1} \in \arg\min\nolimits_{u \in \mathbb{R}^k}\{\tilde{H}\big(X_t^k, t; u, p_t^k\big)\}, \qquad \text{where } \tilde{H} \text{ is the simplified Hamiltonian in (14).} \quad (53)$$

Observe that this algorithm faces two obstacles in high dimensions. First, Step 2 is not fully specified: computing $q^k$ requires solving the BSDE $dp_t^k = -\nabla_x \mathcal{H}\big(X_t^k, t; u_t^k, p_t^k, q_t^k\big)\,\mathrm{d}t + q_t^k\,\mathrm{d}B_t$, which the SMP does not prescribe how to do. This requirement remains even when $\sigma$ does not depend on $u$ or $x$, since $q_t^k$ still appears in the stochastic term $q_t^k\,\mathrm{d}B_t$. In principle, one could approximate $q_t^k$ by solving the full high-dimensional BSDE using deep learning–based methods such as the Deep BSDE approach E et al. (2017); Han et al. (2018) (see Han et al. (2024) for a review), but this incurs substantial additional cost. Alternatively, using (26), one could construct $q_t^k = A(t; X^k)\sigma(t)$, where $A(t; X^k)$ is the unbiased estimate of $\nabla_x^2 J(u^k; X_t^k, t)$ from the matrix SDE equation (32); this is also significantly more expensive because it propagates a $d \times d$ matrix SDE. Second, while the Hamiltonian minimization in Step 3 is well defined and may even admit an explicit solution in function space, this does not directly translate into an implementable update once the control is represented by a neural network; one must instead rely on an optimization procedure in parameter space. The following theorem resolves both issues at once: the gradient flow of the lean AM loss coincides with a continuous-time MSA that directly optimizes the Hamiltonian of Step 3, while avoiding any explicit computation of the adapted process $q_t$ required in Step 2.

**Theorem 5** (Adjoint matching as the continuous-time MSA). *Consider the following continuous-time variant of the MSA where we parameterize the control with a vector field: $u_t = u(X_t^u, t)$, and we consider the following dynamics for the vector field $u$ indexed by a continuous variable $\tau$ instead of $k$:*

$$\begin{cases} \mathrm{d}X_t^\tau = b(X_t^\tau, u^\tau(X_t^\tau, t), t)\mathrm{d}t + \sigma(t)\mathrm{d}B_t, & X_0^\tau \sim P_0, & (54) \\ \mathrm{d}p_t^\tau = -\big(\nabla_1 f(X_t^\tau, u^\tau(X_t^\tau, t), t) + \nabla_1 b(X_t^\tau, u^\tau(X_t^\tau, t), t)^\top p_t^\tau\big)\,\mathrm{d}t + q_t^\tau\,\mathrm{d}B_t, & p_T^\tau = \nabla_x g(X_T^\tau), & (55) \\ \frac{\mathrm{d}}{\mathrm{d}\tau}u^\tau = -\frac{\mathrm{d}}{\mathrm{d}u}\mathbb{E}\big[\int_0^T \tilde{H}(X_t^\tau, t; u(X_t^\tau, t), p_t^\tau)\,\mathrm{d}t\big]|_{u=u^\tau}, & & (56) \end{cases}$$

*where the last line is understood in the formal functional-derivative sense on the control space. With this interpretation, the variational update induced by the expected lean adjoint matching loss coincides with that induced by the continuous-time MSA objective:*

$$\frac{\mathrm{d}}{\mathrm{d}\tau}u^\tau = -\frac{\mathrm{d}}{\mathrm{d}u}\mathbb{E}\big[\mathcal{L}_{\mathrm{AM}}(u; X^{\bar{u}})\big]|_{u=u^\tau}. \quad (57)$$

In other words, each gradient step of the lean AM loss can be viewed as an implementable surrogate for one idealized MSA improvement step: it simulates the forward SDE, uses the corresponding adjoint BSDE $p_t^\tau$ as the conceptual SMP reference object, and improves the control by descending the simplified Hamiltonian $\tilde{H}$ rather than by carrying out the full control-space minimization in the original MSA. The key observation, detailed in the proof, is that although $p_t^\tau$ in equation (55) is coupled to the unknown $q_t^\tau$ through the adjoint BSDE, the lean adjoint $\tilde{a}$ still yields an unbiased estimator of the Hamiltonian gradient with respect to the control, and the gradient flow therefore recovers the MSA control-improvement direction without ever requiring access to $q_t$, resolving the obstacle identified at the end of Section 2.4.

This result can be viewed as the stochastic counterpart of an observation by Li et al. (2018) in the deterministic setting, where they show that gradient descent with backpropagation is equivalent to a discrete-time MSA with the Hamiltonian maximization replaced by a single gradient step. Theorem 5 extends this correspondence to the stochastic regime, where the lean adjoint bypasses the martingale term $q_t$ in the SMP adjoint BSDE that would otherwise need to be solved explicitly. More broadly, this suggests that the lean adjoint construction may serve as a practical alternative to full BSDE solvers, opening a route to MSA-type algorithms for stochastic control beyond generative modeling.

**Remark 6** (Trust-region perspective)**.** *Interpreting the lean adjoint matching as a continuous-time version of MSA is useful beyond the proof itself. It places the lean adjoint matching within a well-studied control-theoretic framework and suggests concrete directions for more stable algorithm design. In the deterministic setting, Li et al. (2018) show that the basic MSA can diverge when the Hamiltonian maximization step induces large "feasibility errors" in the forward–backward dynamics, and propose an extended MSA that regularizes the control update to prevent this. This analysis carries a clear message for the lean adjoint matching in the stochastic setting: updates should be moderated when the iterate is still far from a self-consistent forward–backward pair. This viewpoint naturally motivates trust-region style variants of the lean adjoint matching, where each step is restricted to stay within a neighborhood of the current iterate, improving robustness and preventing divergence, as recently explored in Blessing et al. (2025). Developing analogous convergence guarantees for the lean adjoint matching, informed by the error estimates of Li et al. (2018), is an interesting direction for future work.*

## 5 Numerical Results

In this section, we numerically examine the general adjoint-matching framework developed above. Motivated by recent generative models based on non-Gaussian, state-dependent stochastic processes (Richemond et al., 2022; Avdeyev et al., 2023; Shetty et al., 2025; Kim et al., 2025; Tang et al., 2026), we consider two target-steering problems with geometric Brownian motion (GBM) as the common base process. Both use a density-ratio terminal cost, a common construction in stochastic-control formulations of sampling that expresses the desired terminal law relative to the uncontrolled law (Berner et al., 2023; Havens et al., 2025; Liu et al., 2025).

Within this shared construction, the first experiment is synthetic: it prescribes a three-component Gaussian-mixture target for a correlated GBM and studies performance across increasing ambient dimensions. The second experiment prescribes the distribution of a single MNIST digit in the latent space of a VAE, while decoded terminal samples provide a qualitative image-space assessment. Both settings admit a tractable exact solution. In both settings, the diffusion depends on the state but not on the control, permitting a direct examination of the state-dependent terms in the first-order BAM adjoint and comparison with the lean AM specialization.

### 5.1 Shared controlled-GBM formulation

In both experiments, the controlled state is a positive GBM state $X_t \in \mathbb{R}^d_{>0}$. A Markov control $u : \mathbb{R}^d_{>0} \times [0, T] \to \mathbb{R}^d$ changes the componentwise log-growth rate, giving the controlled dynamics

$$\mathrm{d}X_t = \mathrm{Diag}(X_t)\left(u(X_t, t) + \tfrac{1}{2}\mathrm{diag}(D(t))\right)\mathrm{d}t + \mathrm{Diag}(X_t)S(t)\,\mathrm{d}B_t, \qquad D(t) := S(t)S(t)^\top. \tag{58}$$

Here, $\mathrm{Diag}(x)$ denotes the diagonal matrix with diagonal $x$, while $\mathrm{diag}(D)$ denotes the vector of diagonal entries of $D$. We minimize the path-integral control objective

$$\mathcal{J}(u) = \mathbb{E}\left[\int_0^T \tfrac{1}{2}u(X_t, t)^\top R(t)u(X_t, t)\,\mathrm{d}t + g(X_T)\right], \qquad R(t) = \lambda D(t)^{-1}. \tag{59}$$

The diffusion in equation (58) is state-dependent: $\sigma(x, t) = \mathrm{Diag}(x)S(t)$, so $\sigma_{\cdot k}(x, t) = \mathrm{Diag}(x)S_{\cdot k}(t)$ and $G_k^u(x, t) = \nabla_x \sigma_{\cdot k}(x, t) = \mathrm{Diag}(S_{\cdot k}(t)) \neq 0$. Thus the first-order BAM adjoint in equation (30) retains the drift and diffusion terms involving $G_k^u$, including its stochastic component, whereas lean AM corresponds to the simplification $G_k^u = 0$. Since $\sigma$ is independent of $u$, the Hessian term in the Hamiltonian is constant

with respect to the control. The resulting BAM update is therefore first-order and directly implementable; no $d \times d$ second-order adjoint matrix is simulated in either experiment.

The target-steering construction is shared by the two experiments. Both start from a fixed state. Let $\rho_T^0$ denote the uncontrolled terminal density of $X_T$, let $\rho_T$ be a prescribed target density on $\mathbb{R}_{>0}^d$, and let $\rho_T^*$ denote the terminal density induced by the optimal control. For the objective in equation (59), the path-integral identity (Kappen, 2005; Domingo-Enrich et al., 2024) gives

$$\rho_T^*(x) = \frac{\rho_T^0(x)e^{-g(x)/\lambda}}{\int \rho_T^0(\tilde{x})e^{-g(\tilde{x})/\lambda}\,\mathrm{d}\tilde{x}}. \tag{60}$$

We therefore choose the density-ratio terminal cost

$$g(x) = \lambda\left(\log \rho_T^0(x) - \log \rho_T(x)\right), \qquad \text{which yields} \qquad \rho_T^*(x) = \rho_T(x). \tag{61}$$

For this choice, the desirability function $\Psi$ and exact optimal control are given by

$$\Psi(t,x) := \mathbb{E}^0\left[\frac{\rho_T(X_T)}{\rho_T^0(X_T)} \mid X_t = x\right], \qquad u^*(x,t) = D(t)\operatorname{Diag}(x)\nabla_x \log \Psi(t,x), \tag{62}$$

where the expectation is under the uncontrolled GBM. Its terminal law is lognormal, equivalently Gaussian after the log transform, and the target choices below make $\Psi$ and $u^*$ available in closed form. BAM and lean AM use the same simulated paths, policy class, discretization, and fitting procedure; they differ only in the first-order adjoint target. The shared derivation, precise adjoint equations, and implementation details are given in Section A.

## 5.2 High-dimensional correlated GBM

We first consider a correlated GBM equation (58) with $S(t) \equiv S$. For the target construction, let $y = \log x$ and $z = (y_1, y_2)$. Panel (A) of Figure 1 displays terminal samples in this first two-dimensional log-coordinate plane. We prescribe the terminal marginal density of $z$ as the equally weighted three-component Gaussian mixture

$$q_{T,z}(z) = \tfrac{1}{3}\sum_{j=1}^3 \mathcal{N}(z;\mu_j,\Sigma_j). \tag{63}$$

The three stars in panel (A) mark the component means $\mu_1, \mu_2, \mu_3$. We extend this marginal to a full $d$-dimensional target by retaining the uncontrolled conditional law of the remaining log coordinates given $z$; the precise construction is given in Sections A.1 and A.2. Although the density-ratio terminal cost then depends only on the two displayed coordinates, the off-diagonal correlations in $D$ make the exact optimal control nonzero in all $d$ coordinates. We vary $d \in \{2, 5, 10, 20\}$ while holding the two-dimensional target fixed.

Panel (A) of Figure 1 illustrates the terminal distributions for $d = 20$: lean AM assigns essentially zero mass to one target component, whereas BAM retains nontrivial mass in all three. Lean AM loses the same target mode at every tested dimension. Panel (B) quantifies this discrepancy and shows that BAM attains both a more accurate terminal mode allocation and a lower policy cost throughout the dimension sweep. Detailed metrics are reported in Table 2.

## 5.3 MNIST VAE geometric latent bridge

We next apply the same controlled GBM in a 16-dimensional coordinate system derived from the latent space of a convolutional VAE for MNIST. A scaled, coordinatewise-whitened VAE latent code is used as the GBM log-state $y = \log x$. Every path starts from the fixed all-digit latent mean. The uncontrolled terminal law is therefore Gaussian in these whitened latent coordinates. Applying the inverse whitening map followed by the VAE decoder produces the mixed all-digit base samples shown in the first row of panel (A) of Figure 2. The experiment steers this latent distribution toward a target digit distribution. The precise base process is given in Section A.3.

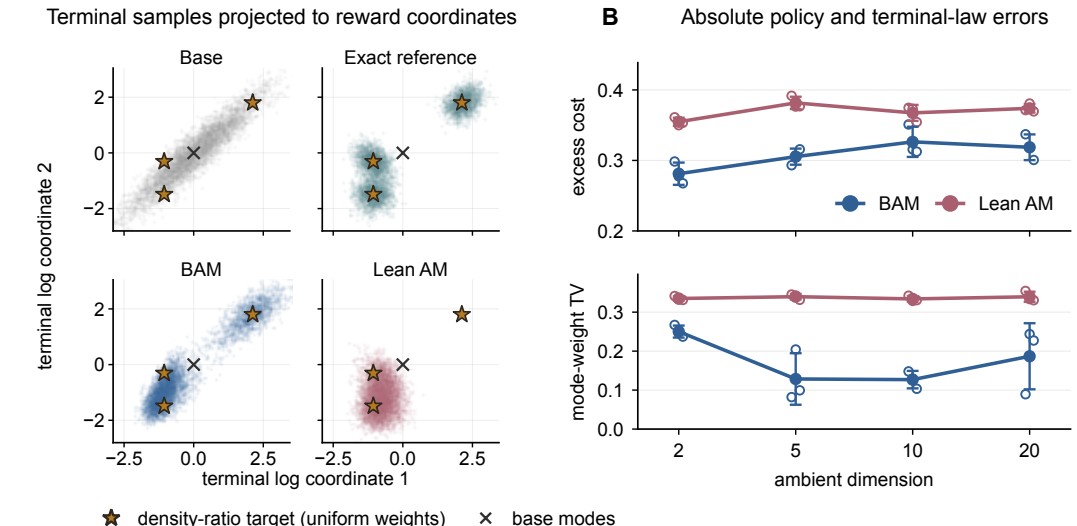

Figure 1: **High-dimensional correlated GBM with a prescribed three-mode terminal law. A:**
Terminal samples for $d = 20$, projected onto the two target log coordinates. Stars mark the three equally
weighted target-component centers and the cross marks the mode of the uncontrolled terminal distribution.
BAM retains all three target modes, whereas lean AM loses one mode and misallocates mass between the
other two. **B:** Excess policy cost above the exact reference (top) and total-variation distance between the
terminal mode weights and the direct-target reference weights (bottom) for $d \in \{2, 5, 10, 20\}$. Open circles
show three individual seeds; filled circles and error bars show mean $\pm$ standard deviation.

We steer the base distribution toward one selected digit by prescribing the terminal density in the GBM log
coordinates as an equally weighted Gaussian mixture

$$q_T(y) = \tfrac{1}{M} \sum_{j=1}^{M} \mathcal{N}(y; c_j, \Sigma_{\text{tgt}}), \tag{64}$$

where the centers $c_j$ are encoded examples of that digit and $M = 96$. We use digit 5 in the main text
and report results for additional single-digit targets in Section A.4. This is the same density-ratio problem
as the synthetic experiment, with a higher-dimensional, data-derived mixture replacing equation (63). The
primary evaluation remains in latent space, where terminal sliced-Wasserstein distance to $q_T$ and control
error to $u^*$ are calibrated against known references. Decoded samples and LeNet feature distance provide
complementary image-space diagnostics but are not part of the controlled objective.

Panel (A) of Figure 2 shows decoded samples from the uncontrolled base, exact target, BAM, and lean AM
terminal laws. BAM produces more consistently recognizable digit-5 samples, whereas lean AM remains more
dispersed. Panel (B) quantifies the comparison: BAM is closer than lean AM to the exact target in latent
sliced-Wasserstein distance and has a lower LeNet-FID after decoding. Because the decoded samples and
LeNet features also depend on the fixed VAE decoder and classifier, we treat these image-space comparisons
as supporting diagnostics rather than the primary evidence. The comparison with the exact control, detailed
metrics, and results for additional target digits are reported in Table 3 and section A.4.

Taken together, the experiments show the same effect for two prescribed Gaussian-mixture targets: an
analytically transparent three-mode law and a data-derived law in VAE latent space, with decoding used only
for image-space diagnostics. For state-dependent but control-independent diffusion, retaining the diffusion-
Jacobian terms in the first-order BAM adjoint improves the attained objective and terminal distribution
in both settings, and substantially reduces control error in the MNIST experiment. These results provide
numerical evidence for the correctness and practical relevance of the BAM construction.

## 6 Related work

Figure 2: **MNIST VAE geometric latent bridge for target digit** 5. **A:** Randomly selected decoded terminal samples from the uncontrolled base distribution, the exact target distribution, BAM, and lean AM. **B:** Terminal sliced-Wasserstein distance in latent space and LeNet feature Fréchet distance (FID) to samples from the exact target. Bars and error bars show mean ± standard deviation over three seeds; open circles show individual seeds. Lower is better.

Beyond the original adjoint matching paper of Domingo-Enrich et al. (2025), which our results revisit and generalize, a rapidly growing family of methods builds on the lean adjoint matching loss in increasingly broad settings.

**Sampling-oriented descendants of adjoint matching.** Havens et al. (2025) introduced *Adjoint Sampling*, which uses lean adjoint matching together with a memoryless reference SDE to train scalable diffusion samplers for unnormalized target densities, with applications to molecular conformer generation. Liu et al. (2025) extended this idea to Schrödinger-bridge dynamics in their *Adjoint Schrödinger Bridge Sampler*, which trades the requirement of a memoryless prior for an additional learned base process. Park et al. (2025) further extend the approach to infinite-dimensional state spaces (*Functional Adjoint Sampler*), targeting sampling problems on function spaces. A central practical advantage of the lean formulation in these settings is that, in the time-dependent-diffusion / control-affine regime, the lean adjoint can be evaluated *simulation-free* by renoising samples along a known reference SDE, which dramatically improves sample efficiency and scalability. Our analysis applies directly to those settings and gives a control-theoretic explanation for the tractability of the lean loss.

**Discrete-state extensions.** So et al. (2026) propose *Discrete Adjoint Matching*, extending the lean adjoint matching framework to discrete-state diffusion models, and Guo et al. (2026) develop a corresponding *Discrete Adjoint Schrödinger Bridge Sampler*. Relatedly, Zhu et al. (2025) formulate masked diffusion neural samplers (*MDNS*) as discrete stochastic optimal control problems and solve them via adjoint-based losses. The Hamiltonian viewpoint developed here suggests an analogous route beyond continuous state spaces, but the explicit SMP machinery we use relies on Brownian dynamics, gradients, Itô calculus, and BSDEs. A rigorous discrete analogue would therefore require the corresponding discrete-state optimal-control machinery, and we leave this extension as future work.

**Related sampling and fine-tuning frameworks.** Domingo-Enrich et al. (2026) provide a unified perspective on fine-tuning and sampling with diffusion and flow models, relating adjoint-matching-style losses to other stochastic control–based losses for generative models. Ren et al. (2026) propose *DriftLite*, a lightweight inference-time drift-control method for scaling diffusion models, which uses control ideas similar in spirit to ours but at inference rather than at training time. From a different direction, Li & Levine (2026) use adjoint matching as a building block for off-policy reinforcement learning in robotics (*Q-learning with Adjoint Matching*), illustrating that the algorithmic ideas connecting adjoint matching to the SMP have impact well beyond generative modeling.

**Stochastic control side.** On the stochastic control side, our results are closest to the method of successive approximations for SOC (Chernousko & Lyubushin, 1982; Li et al., 2018; Kerimkulov et al., 2021) and to learning-based approaches for solving SMP adjoint BSDEs (E et al., 2017; Han et al., 2018; 2024). These methods aim to make maximum-principle-based control algorithms implementable despite the martingale integrand in the stochastic adjoint equation. Theorem 5 gives a complementary route in the time-dependent-diffusion setting: the lean AM gradient flow realizes the corresponding continuous-time MSA direction without explicitly solving for that martingale integrand.

## 7 Conclusion

In this work, we revisited adjoint matching from the perspective of stochastic optimal control (SOC) and clarified its structure at three levels. At the most general level, we introduced a Hamiltonian basic adjoint matching (BAM) objective whose expected first variation matches that of the original SOC problem. In the practically important case of purely time-dependent diffusion, this reduces to a lean adjoint matching (AM) objective that avoids second-order quantities while retaining the correct critical points under the corresponding verification assumptions. Finally, through the stochastic maximum principle (SMP), we showed how the lean adjoint matching can be interpreted as a principled continuous-time method of successive approximations (MSA)-type update that avoids explicit computation of the martingale term in the SMP adjoint backward stochastic differential equation (BSDE). This perspective provides a principled foundation for the adjoint matching framework and suggests a systematic route for designing scalable SOC-inspired algorithms in generative modeling. From a stochastic control standpoint, our results contribute new implementable objectives for general SOC whose critical points satisfy the Hamilton–Jacobi–Bellman (HJB) stationarity conditions, together with a practical pathway for MSA-type iterations that avoids solving BSDEs. We hope this encourages further exchange between the stochastic control and generative modeling communities.

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

# A    Numerical experiment details

This appendix gives the common analytic reference, numerical updates, evaluation metrics, and experiment-specific settings used for Section 5. Both experiments use the controlled GBM in equation (58) and start from the deterministic source $Y_0 = \mathbf{0}$, equivalently $X_0 = \mathbf{1}$; they differ only in the prescribed terminal mixture and in the feature representation used for policy fitting.

## A.1    Shared controlled-GBM and path-integral reference

For the analytic calculations, let $Y_t = \log X_t$ componentwise and write $\bar{u}(y,t) = u(e^y, t)$. Itô's formula applied to equation (58) gives the additive-noise process

$$\mathrm{d}Y_t = \bar{u}(Y_t,t)\,\mathrm{d}t + S(t)\,\mathrm{d}B_t. \tag{65}$$

The uncontrolled transition $Y_T \mid Y_t = y$ is therefore Gaussian with mean $y$ and covariance $C_{t,T} := \int_t^T D(s)\,\mathrm{d}s$. Let $p_T^0$ and $q_T$ denote the uncontrolled and target terminal densities in the $Y$-coordinates. The corresponding $X$-space densities satisfy

$$\rho_T^0(x) = \frac{p_T^0(\log x)}{\prod_{i=1}^d x_i}, \qquad \rho_T(x) = \frac{q_T(\log x)}{\prod_{i=1}^d x_i}, \qquad \frac{\rho_T(x)}{\rho_T^0(x)} = \frac{q_T(\log x)}{p_T^0(\log x)}. \tag{66}$$

Thus the density-ratio cost in equation (61) can equivalently be written as $G(y) := g(e^y) = \lambda\{\log p_T^0(y) - \log q_T(y)\}$. The desirability function and exact control become

$$\psi(t,y) = \mathbb{E}^0\left[\frac{q_T(Y_T)}{p_T^0(Y_T)} \mid Y_t = y\right], \tag{67}$$

$$\bar{u}^*(y,t) = D(t)\nabla_y \log \psi(t,y), \qquad u^*(x,t) = \bar{u}^*(\log x, t). \tag{68}$$

In both experiments, $p_T^0$ is Gaussian and the target is a finite Gaussian mixture $q_T(y) = \sum_{j=1}^M \pi_j \mathcal{N}(y; c_j, \Sigma_j)$. Consequently,

$$\psi(t,y) = \sum_{j=1}^M \pi_j \int_{\mathbb{R}^d} \mathcal{N}(v; y, C_{t,T}) \frac{\mathcal{N}(v; c_j, \Sigma_j)}{p_T^0(v)}\,\mathrm{d}v. \tag{69}$$

Each summand is an integral of the exponential of a quadratic form and is evaluated analytically by completing the square. Differentiating the resulting finite sum gives $\nabla_y \log \psi$ and hence the exact control in equation (67). Exact target samples are drawn directly from $q_T$, avoiding time-discretization error in the terminal reference distribution.

### A.1.1    Implemented BAM and lean AM updates

We use the uniform grid $t_n = n\Delta t$, with $\Delta t = T/N$, and simulate equation (65) by

$$Y_{n+1} = Y_n + \bar{u}(Y_n, t_n)\Delta t + S(t_n)\Delta B_n, \qquad X_{n+1} = e^{Y_{n+1}}. \tag{70}$$

Let $a_n$ denote the discrete-time analog of the continuous first-order adjoint $a_t$ in equation (30), evaluated at the grid points $t_n$, and define the log-coordinate adjoint $r_n := X_n \odot a_n$, where $\odot$ denotes componentwise multiplication. The two parameterizations are exactly equivalent; we discretize in log coordinates because the diffusion $\sigma(x,t) = \mathrm{Diag}(x)S(t)$ is state-dependent in $X$ but state-independent after the log-transform, which yields a stable recursion without an explicit $G_k^u$ term. If $J_n := \nabla_y \bar{u}(Y_n, t_n)$, the backward BAM recursion used in the experiments is

$$r_N = \nabla_y G(Y_N), \tag{71}$$

$$r_n = r_{n+1} + \Delta t\, J_n^\top (r_{n+1} + R(t_n)\bar{u}(Y_n, t_n)), \qquad a_n = X_n^{-1} \odot r_n. \tag{72}$$

This recursion is the GBM specialization of the full first-order adjoint equation (30); the state-dependence of $\sigma$ is fully retained, encoded compactly through the log-coordinate drift $\bar{u}$ rather than through the explicit $G_k^u$

terms of the $X$-coordinate form. Lean AM instead targets the lean adjoint equation (9) of Domingo-Enrich et al. (2025), which is the $G_k^u \equiv 0$ simplification of equation (30) appropriate when $\sigma(x, u, t) := \sigma(t)$. It discretizes the lean adjoint directly in the $X$-coordinates, giving

$$a_N = \nabla_x g(X_N), \tag{73}$$

$$a_n = a_{n+1} \odot \exp\left[\left(\bar{u}(Y_n, t_n) + \tfrac{1}{2}\operatorname{diag}(D(t_n))\right)\Delta t\right], \tag{74}$$

where the exponential is componentwise. The BAM and lean AM recursions, equation (71) and equation (73), use the same forward simulator but target two distinct adjoints: BAM the full first-order adjoint equation (30), lean AM the lean adjoint equation (9). Since the diffusion is independent of the control, the second-order Hamiltonian term does not affect the control update, and neither method propagates the general $d \times d$ second-order adjoint matrix.

Two technical points are worth noting. First, the continuous adjoint equation (30) carries a $\mathrm{d}B_t$ term while the discrete recursion equation (71) does not. For GBM, Itô's product formula applied to $r_t = X_t \odot a_t$ gives, componentwise, a martingale coefficient $a_t^{(i)} X_t^{(i)} S_{ik}(t) - X_t^{(i)} a_t^{(i)} S_{ik}(t) = 0$ for each Brownian coordinate $k$. Thus $r_t$ still depends on the realized forward trajectory, but it has no martingale increment once that trajectory is fixed, and we discretize this pathwise finite-variation equation directly. Second, the state-coordinate representation must be separated from the adjoint approximation. Rewriting the lean recursion for the $X$-state-coordinate adjoint $a_n$ in terms of $r_n = X_n \odot a_n$ gives the equivalent update

$$r_n = r_{n+1} \odot \exp\left[\frac{1}{2}\operatorname{diag}(D(t_n))\Delta t - S(t_n)\Delta B_n\right],$$

which still contains the forward noise increment. This is different from applying the lean simplification after the log transform: in $Y$-coordinates the diffusion is $\bar{\sigma}(y, t) = S(t)$, so $\nabla_y \bar{\sigma}_{\cdot k} \equiv 0$, and setting $G_k^u \equiv 0$ there would recover the full log-coordinate adjoint, namely the BAM recursion equation (71). We therefore present BAM in the log-coordinate adjoint representation natural for the full adjoint and lean AM through the $X$-state-coordinate adjoint used by Domingo-Enrich et al. (2025).

For either adjoint, Hamiltonian stationarity gives the pathwise policy target

$$\widehat{u}_n = -R(t_n)^{-1} r_n. \tag{75}$$

For lean AM, $r_n$ in equation (75) denotes $X_n \odot a_n$ after computing the $X$-coordinate recursion equation (73). At each time step, the policy is linear in a feature vector $\phi(y)$ containing a constant, all linear coordinates of $y$, and Gaussian radial basis function (RBF) features. We fit $\widehat{u}_n$ by ridge regression and damp the fitted update:

$$\widehat{W}_n = \arg\min_W \sum_{\ell=1}^{n_{\mathrm{train}}} \left\| W^\top \phi(Y_n^{(\ell)}) - \widehat{u}_n^{(\ell)} \right\|_2^2 + \gamma\|W\|_{\mathrm{F}}^2, \tag{76}$$

$$W_n^{k+1} = (1 - \eta)W_n^k + \eta\widehat{W}_n, \tag{77}$$

where the superscript $k$ is again the index of outer loop iteration in MSA, as a stable implementation of adjoint matching. BAM and lean AM use the same feature class, path budget, simulation seeds, and final-iterate evaluation; only the adjoint they target differs. Before each regression in both experiments, training trajectories with nonfinite values, nonpositive states, or exceptionally large states or adjoints are excluded; in the retained runs this filter acts only as a rare-outlier safeguard.

### A.1.2 Metrics

For a fitted policy $\widehat{u}$, the excess policy cost is $\mathcal{J}(\widehat{u}) - \mathcal{J}(u^*)$. Relative control error is evaluated along paths generated by $\widehat{u}$ and discretized on the same time grid:

$$\mathrm{Err}_u = \left(\frac{\sum_{n,\ell} \|\widehat{u}(X_n^{(\ell)}, t_n) - u^*(X_n^{(\ell)}, t_n)\|_2^2}{\sum_{n,\ell} \|u^*(X_n^{(\ell)}, t_n)\|_2^2}\right)^{1/2}. \tag{78}$$

For the synthetic experiment, each terminal sample is assigned to its nearest target center and the mode-weight error is the total-variation distance between the resulting empirical weights and those obtained from direct target samples. For MNIST, terminal sliced-Wasserstein distance is computed in the 16-dimensional log-latent coordinates using 64 random one-dimensional projections and direct samples from $q_T$. LeNet-FID is computed after decoding, using the frozen LeNet penultimate-layer features. All reported entries are means and sample standard deviations over three independent seeds; lower is better for every metric.

## A.2 High-dimensional correlated GBM

The synthetic experiment uses constant diffusion $S(t) \equiv S$, and partitions log coordinates into active indices $A = \{1, 2\}$ and inactive indices $I = \{3, \ldots, d\}$. The uncontrolled terminal law is then $p_T^0 = \mathcal{N}(\mathbf{0}, TSS^\top)$, and the prescribed target extending the active marginal mixture in equation (63) to all $d$ coordinates is

$$q_T(y) = \left[ \frac{1}{3} \sum_{j=1}^3 \mathcal{N}(y_A; \mu_j, \Sigma_j) \right] p_T^0(y_I \mid y_A). \tag{79}$$

with three equally weighted active components placed around the origin in the active subspace (Figure 1, panel A).

$S$ is a fixed structured matrix with a $2 \times 2$ active block and inactive couplings into the active block. The retained runs use $d \in \{2, 5, 10, 20\}$, $T = 1$, $\lambda = 0.3$, $N = 60$ time steps, 120 policy updates, 800 training paths per update, 5000 evaluation paths, damping $\eta = 0.01$, and ridge parameter $\gamma = 3 \times 10^{-4}$. The RBF features are centered at the uncontrolled active mean and the three active target centers with Gaussian bandwidth 0.85, while the linear block includes all $d$ log coordinates. Panel (A) of Figure 1 shows a representative seed at $d = 20$.

Table 2: High-dimensional correlated-GBM metrics. Entries are mean $\pm$ standard deviation over three seeds. TV denotes the total-variation distance between terminal mode weights and the direct-target reference weights.

| $d$ | Excess policy cost ↓ | | Mode-weight TV ↓ | | Relative control error ↓ | |
|---|---|---|---|---|---|---|
| | BAM | lean AM | BAM | lean AM | BAM | lean AM |
| 2 | $\mathbf{0.281 \pm 0.016}$ | $0.355 \pm 0.006$ | $\mathbf{0.250 \pm 0.015}$ | $0.335 \pm 0.006$ | $\mathbf{0.718 \pm 0.015}$ | $0.730 \pm 0.006$ |
| 5 | $\mathbf{0.305 \pm 0.011}$ | $0.382 \pm 0.009$ | $\mathbf{0.129 \pm 0.066}$ | $0.340 \pm 0.007$ | $\mathbf{0.719 \pm 0.043}$ | $0.736 \pm 0.004$ |
| 10 | $\mathbf{0.326 \pm 0.021}$ | $0.367 \pm 0.011$ | $\mathbf{0.127 \pm 0.022}$ | $0.334 \pm 0.007$ | $0.741 \pm 0.010$ | $\mathbf{0.738 \pm 0.008}$ |
| 20 | $\mathbf{0.319 \pm 0.018}$ | $0.374 \pm 0.006$ | $\mathbf{0.187 \pm 0.085}$ | $0.339 \pm 0.013$ | $0.754 \pm 0.019$ | $\mathbf{0.747 \pm 0.003}$ |

## A.3 MNIST VAE geometric latent bridge

A convolutional VAE trained on 30000 MNIST images maps each image to a latent mean $\zeta \in \mathbb{R}^{16}$. Let $\mu_\zeta$ and $s_\zeta$ be the coordinatewise mean and standard deviation of $\zeta$ over the encoded MNIST bank. We define a GBM state $x \in \mathbb{R}_{>0}^{16}$ from $\zeta$ by

$$y = \alpha(\zeta - \mu_\zeta) \oslash s_\zeta, \qquad \alpha = 0.5, \qquad x = e^y, \tag{80}$$

where $\alpha$ keeps $x$ within a numerically moderate range and $y = \log x$ is the auxiliary log-coordinate as in equation (65).

The initial state is the all-digit latent mean $\mu_\zeta$, which equation (80) maps to $y = \mathbf{0}$ and $X_0 = \mathbf{1}$. The diffusion has the form $S(t) = s(t)S_0$, where the scalar schedule $s(t) = 0.01 + 1.99t^{3/2}$ ramps the noise level over time and $S_0$ is a fixed lower-triangular matrix with correlated off-diagonal entries. The uncontrolled terminal log law is therefore $p_T^0 = \mathcal{N}(\mathbf{0}, \int_0^T S(t)S(t)^\top \, dt)$.

For a selected target digit, we encode $M = 96$ MNIST examples of that digit, map them through equation (80) to obtain centers $c_j \in \mathbb{R}^{16}$ in the $y$-coordinate, and prescribe the Gaussian-mixture target

$$q_T(y) = \frac{1}{M} \sum_{j=1}^{M} \mathcal{N}(y; c_j, \Sigma_{\text{tgt}}), \tag{81}$$

$$\Sigma_{\text{tgt}} = h^2 \operatorname{Diag} \left( \max\{\widehat{\operatorname{Var}}_{\text{target}}(y_i), \, 0.05^2\} \right)_{i=1}^{16}, \qquad h = 0.60. \tag{82}$$

The mixture covariance follows the coordinatewise spread of the selected digit rather than imposing a common isotropic scale, and the density-ratio terminal cost admits the same closed-form reference equation (69) as the synthetic experiment. Terminal $X_T$ samples are inverted back to $\zeta_T$ via equation (80) and decoded by the VAE for image-space diagnostics.

The retained runs use $T = 1$, $\lambda = 0.5$, $N = 40$ time steps, 800 training paths, 1000 evaluation paths, 60 policy updates, and ridge parameter $10^{-6}$. The RBF policy uses 64 sampled feature centers together with the 96 target centers, for 160 RBF centers in total, with Gaussian bandwidth 2.2. BAM uses damping 0.05 and lean AM uses damping 0.007.

The latent sliced-Wasserstein and relative-control errors are the primary diagnostics because they compare directly with the known target and exact control; decoded samples and LeNet-FID are supporting, since they also reflect the fixed VAE decoder and classifier, neither of which appears in the control objective.

### A.4  Additional MNIST target digits

Target digits 7 and 9 provide additional single-digit steering examples under the same setup as digit 5. Only the encoded examples used to construct the target mixture in equation (81) change with the target digit; all other settings, including the initial state and base uncontrolled process, are reused.

Table 3: MNIST VAE latent-bridge metrics. Entries are mean $\pm$ standard deviation over three seeds. SW is the terminal sliced-Wasserstein distance to the exact latent target; FID is computed from decoded samples using the frozen LeNet features.

| Target digit | Terminal SW ↓ | | Relative control error ↓ | | LeNet-FID ↓ | |
|---|---|---|---|---|---|---|
| | BAM | lean AM | BAM | lean AM | BAM | lean AM |
| 5 | $\mathbf{0.121 \pm 0.022}$ | $0.532 \pm 0.198$ | $\mathbf{0.370 \pm 0.006}$ | $0.804 \pm 0.010$ | $\mathbf{1.810 \pm 0.028}$ | $17.423 \pm 6.887$ |
| 7 | $\mathbf{0.107 \pm 0.002}$ | $0.461 \pm 0.088$ | $\mathbf{0.305 \pm 0.008}$ | $0.818 \pm 0.013$ | $\mathbf{1.327 \pm 0.142}$ | $8.605 \pm 5.728$ |
| 9 | $\mathbf{0.104 \pm 0.020}$ | $0.442 \pm 0.100$ | $\mathbf{0.305 \pm 0.014}$ | $0.807 \pm 0.007$ | $\mathbf{3.670 \pm 1.013}$ | $13.613 \pm 0.471$ |

For all three target digits, BAM has lower terminal sliced-Wasserstein distance, relative control error, and LeNet-FID than lean AM; the corresponding values are reported in Table 3. Figure 3 shows that the same qualitative and quantitative pattern extends beyond the digit-5 example in the main text.

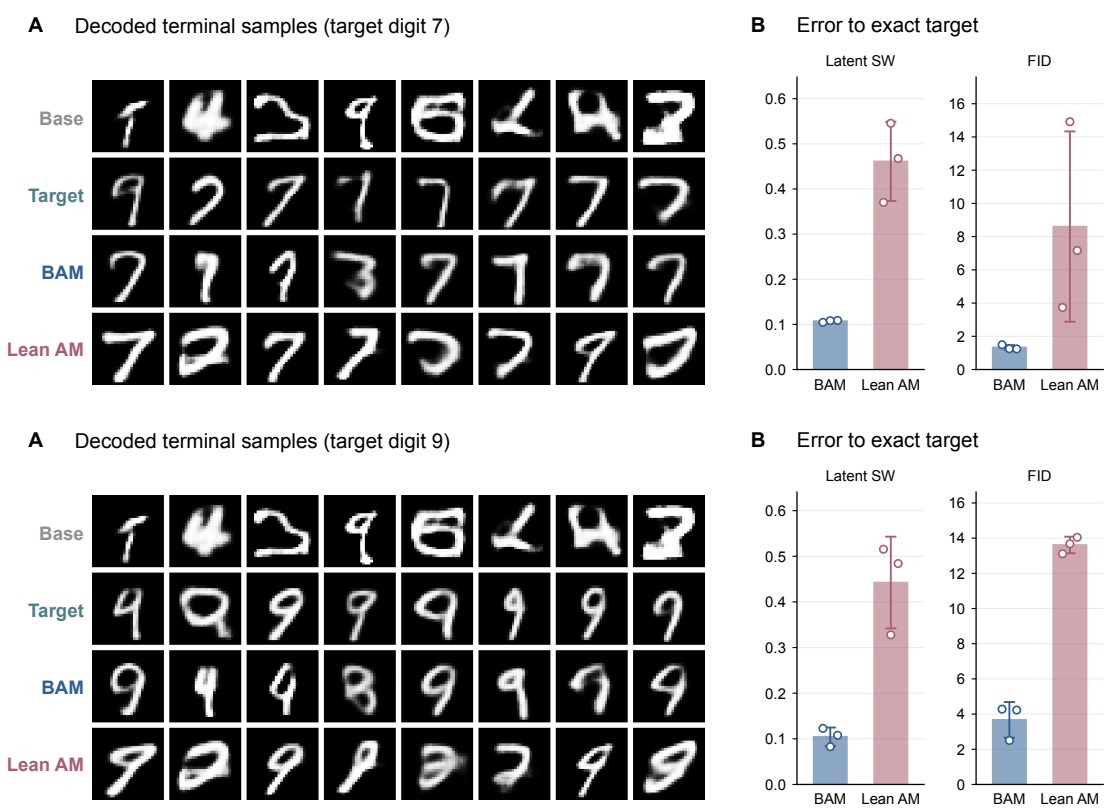

Figure 3: **MNIST VAE geometric latent bridge for target digits 7 (top) and 9 (bottom).** Panel layout and conventions match Figure 2; only the selected target digit and, consequently, the encoded examples that define the target mixture change between rows.

# B  Proof of Theorem 1: Basic Adjoint Matching for General Hamiltonians

Throughout the proofs, we assume $b, \sigma, f$ are $C^1$ in $(x, u)$ and $g$ is $C^1$, all with derivatives of at most polynomial growth; that the controlled SDE equation (4) is well-posed for every admissible control and sufficiently small perturbations thereof, with finite moments of all relevant orders; and that these bounds justify differentiation under the expectation wherever invoked below. When individual results require additional regularity (e.g. $C^2$ for the second-order adjoint, or $J(u; \cdot, \cdot) \in C^{1,2}$ for the verification step), this is stated explicitly.

We write the original control objective as

$$\mathcal{J}(u) := \mathbb{E}\left[\int_0^T f(X_t^u, u(X_t^u, t), t)\, \mathrm{d}t + g(X_T^u)\right], \qquad \mathrm{d}X_t^u = b(X_t^u, u(X_t^u, t), t)\, \mathrm{d}t + \sigma(X_t^u, u(X_t^u, t), t)\, \mathrm{d}B_t. \tag{83}$$

Fix a Markov control $u$ and a perturbation $v$ (same dimension as $u$), and consider $u^\epsilon := u + \epsilon v$ with $\epsilon$ small. By Theorem 7, the standard first-variation identity reads

$$\frac{\mathrm{d}}{\mathrm{d}\epsilon}\mathcal{J}(u^\epsilon)\Big|_{\epsilon=0} = \mathbb{E}\left[\int_0^T \left\langle v(X_t^u, t), \nabla_u H(X_t^u, t; u(X_t^u, t), p_t^u, M_t^u)\right\rangle \mathrm{d}t\right], \tag{84}$$

where the Hamiltonian $H$ is defined in equation (34), and where

$$p_t^u = \nabla_x J(u; X_t^u, t), \qquad M_t^u = \nabla_x^2 J(u; X_t^u, t).$$

Equivalently, one may use any unbiased representation of $p_t^u$ and $M_t^u$ coming from an adjoint/second-adjoint construction, provided it produces the contractions $\langle b, p \rangle$ and $\mathrm{Tr}(\sigma\sigma^\top M)$ inside $H$.

Now we consider the basic adjoint matching loss (36). Then, for the same perturbation $v$, we obtain immediately that

$$\frac{\mathrm{d}}{\mathrm{d}\epsilon}\mathbb{E}[\mathcal{L}_{\mathrm{BAM}}(u^\epsilon; X^{\bar{u}})]\Big|_{\epsilon=0} = \mathbb{E}\left[\int_0^T \left\langle v(X_t^{\bar{u}}, t), \nabla_u H(X_t^{\bar{u}}, t; u(X_t^{\bar{u}}, t), a(t, X^{\bar{u}}), A(t, X^{\bar{u}}))\right\rangle \mathrm{d}t\right]. \tag{85}$$

Observe that by the tower property of expectation,

$$\begin{aligned}
&\mathbb{E}\left[\int_0^T \left\langle v(X_t^{\bar{u}}, t), \nabla_u H(X_t^{\bar{u}}, t; u(X_t^{\bar{u}}, t), a(t, X^{\bar{u}}), A(t, X^{\bar{u}}))\right\rangle \mathrm{d}t\right] \\
&= \mathbb{E}\left[\int_0^T \left\langle v(X_t^{\bar{u}}, t), \nabla_2 f(X_t^{\bar{u}}, u(X_t^{\bar{u}}, t), t) + \nabla_2 b(X_t^{\bar{u}}, u(X_t^{\bar{u}}, t), t)^\top a(t, X^{\bar{u}})\right.\right. \\
&\qquad\qquad \left.\left. + \tfrac{1}{2}\nabla_u \mathrm{Tr}\left(\sigma(X_t^{\bar{u}}, u, t)\sigma(X_t^{\bar{u}}, u, t)^\top A(t, X^{\bar{u}})\right)|_{u=u(X_t^{\bar{u}}, t)}\right\rangle \mathrm{d}t\right] \\
&= \mathbb{E}\left[\int_0^T \left\langle v(X_t^{\bar{u}}, t), \nabla_2 f(X_t^{\bar{u}}, u(X_t^{\bar{u}}, t), t) + \nabla_2 b(X_t^{\bar{u}}, u(X_t^{\bar{u}}, t), t)^\top \mathbb{E}[a(t, X^{\bar{u}})|X_t^{\bar{u}}]\right.\right. \\
&\qquad\qquad \left.\left. + \tfrac{1}{2}\nabla_u \mathrm{Tr}\left(\sigma(X_t^{\bar{u}}, u, t)\sigma(X_t^{\bar{u}}, u, t)^\top \mathbb{E}[A(t, X^{\bar{u}})|X_t^{\bar{u}}]\right)|_{u=u(X_t^{\bar{u}}, t)}\right\rangle \mathrm{d}t\right] \\
&= \mathbb{E}\left[\int_0^T \left\langle v(X_t^{\bar{u}}, t), \nabla_2 f(X_t^{\bar{u}}, u(X_t^{\bar{u}}, t), t) + \nabla_2 b(X_t^{\bar{u}}, u(X_t^{\bar{u}}, t), t)^\top \nabla_x J(\bar{u}; X_t^{\bar{u}}, t)\right.\right. \\
&\qquad\qquad \left.\left. + \tfrac{1}{2}\nabla_u \mathrm{Tr}\left(\sigma(X_t^{\bar{u}}, u, t)\sigma(X_t^{\bar{u}}, u, t)^\top \nabla_x^2 J(\bar{u}; X_t^{\bar{u}}, t)\right)|_{u=u(X_t^{\bar{u}}, t)}\right\rangle \mathrm{d}t\right] \\
&= \mathbb{E}\left[\int_0^T \left\langle v(X_t^{\bar{u}}, t), \nabla_u H(X_t^{\bar{u}}, t; u(X_t^{\bar{u}}, t), p_t^{\bar{u}}, M_t^{\bar{u}})\right\rangle \mathrm{d}t\right]
\end{aligned} \tag{86}$$

Here, the third equality follows from the fact that $\nabla_x J(\bar{u}; x, t) = \mathbb{E}[a(t, X^{\bar{u}})|X_t^{\bar{u}} = x]$ and $\nabla_x^2 J(\bar{u}; x, t) = \mathbb{E}[A(t, X^{\bar{u}})|X_t^{\bar{u}} = x]$ by Lemma 8 and Lemma 9, respectively.

Finally, evaluate the above at $\bar{u} = u$ (the intended usage): since the state process and adjoint objects coincide,

$$(X^{\bar{u}}, p^{\bar{u}}, M^{\bar{u}}) = (X^u, p^u, M^u),$$

we obtain the *same* first variation as in equation (84):

$$\frac{\mathrm{d}}{\mathrm{d}\epsilon}\,\mathbb{E}[\mathcal{L}_{\mathrm{BAM}}(u^\epsilon; X^{\bar{u}})]\Big|_{\epsilon=0,\ \bar{u}=\mathrm{stopgrad}(u)} = \frac{\mathrm{d}}{\mathrm{d}\epsilon}\mathcal{J}(u^\epsilon)\Big|_{\epsilon=0}.$$

This proves the claim that $u \mapsto \mathbb{E}[\mathcal{L}_{\mathrm{BAM}}(u; X^{\bar{u}})]$ has the same first variation as the original objective.

Now let $\hat{u}$ be a critical point of $u \mapsto \mathbb{E}[\mathcal{L}(u; X^{\bar{u}})]$ (with $\bar{u} = \mathrm{stopgrad}(u)$). Then the first variation above vanishes for all perturbations $v$, hence the integrand must vanish (in the usual weak/pointwise sense), yielding the stationarity condition

$$\nabla_u H\big(x, t; \hat{u}(x,t), \nabla_x J(\hat{u}; x, t), \nabla_x^2 J(\hat{u}; x, t)\big) = 0 \quad \text{for a.e. } (x, t), \tag{87}$$

which is equation (37). If, moreover, $u \mapsto H(x, t; u, p, M)$ is convex, this stationarity is equivalent to the pointwise minimizer condition

$$\hat{u}(x, t) \in \arg\min_u H\big(x, t; u, \nabla_x J(\hat{u}; x, t), \nabla_x^2 J(\hat{u}; x, t)\big). \tag{88}$$

By Lemma 10 (the general $f$ verification step), this implies that $J(\hat{u}; \cdot, \cdot)$ satisfies the HJB equation

$$\partial_t J + \inf_u H(x, t; u, \nabla_x J, \nabla_x^2 J) = 0, \qquad J(\cdot, T) = g. \tag{89}$$

If the HJB solution is unique, then $J(\hat{u}; \cdot, \cdot) = V$ and $\hat{u} = u^\star$ is the optimal control.

**Lemma 7** (First variation of $\mathcal{J}$ via adjoints (controlled diffusion))**.** *Fix a Markov control $u$ and a perturbation $v$ of the same dimension, and set $u^\epsilon := u + \epsilon v$. Consider the controlled diffusion*

$$\mathrm{d}X_t^\epsilon = b(X_t^\epsilon, u^\epsilon(X_t^\epsilon, t), t)\,\mathrm{d}t + \sigma(X_t^\epsilon, u^\epsilon(X_t^\epsilon, t), t)\,\mathrm{d}B_t, \qquad X_0^\epsilon \sim P_0,$$

*and the objective*

$$\mathcal{J}(u^\epsilon) = \mathbb{E}\left[ \int_0^T f(X_t^\epsilon, u^\epsilon(X_t^\epsilon, t), t)\,\mathrm{d}t + g(X_T^\epsilon) \right].$$

*Assume $b, \sigma, f$ are $C^1$ in $(x, u)$ with derivatives of at most polynomial growth, $g$ is $C^1$ with at most polynomial growth, and that the SDE is well-posed for all sufficiently small $\epsilon$ with moments ensuring the differentiations below are justified. Assume in addition that, for the frozen control $u$, the associated cost-to-go $J(u; \cdot, \cdot)$ is a classical $C^{1,2}$ solution of the backward equation used in the proof below. Then the directional derivative exists and is given by*

$$\frac{\mathrm{d}}{\mathrm{d}\epsilon}\mathcal{J}(u^\epsilon)\Big|_{\epsilon=0} = \mathbb{E}\left[ \int_0^T \left\langle v(X_t^u, t),\, \nabla_u H\big(X_t^u, t;\, u(X_t^u, t),\, p_t,\, M_t\big) \right\rangle \mathrm{d}t \right], \tag{90}$$

*where $H$ is the Hamiltonian*

$$H(x, t; u, p, M) = f(x, u, t) + \langle b(x, u, t), p \rangle + \tfrac{1}{2}\,\mathrm{Tr}\big(\sigma(x, u, t)\sigma(x, u, t)^\top M\big),$$

*and where $(p_t, M_t)$ are the first- and second-order sensitivity objects*

$$p_t = \nabla_x J(u; X_t^u, t), \qquad M_t = \nabla_x^2 J(u; X_t^u, t).$$

*Equivalently, one may replace $(p_t, M_t)$ by any unbiased adjoint representation that yields the contractions $\langle b, p \rangle$ and $\mathrm{Tr}(\sigma\sigma^\top M)$ appearing in $H$.*

**Lemma 8** (Adjoint method for SDEs with state-dependent diffusion (Domingo-Enrich et al., 2024; Li et al., 2020; Kidger et al., 2021))**.** *Let $(B_t)_{t\in[0,T]}$ be an $m$-dimensional Brownian motion and let $X : \Omega \times [0, T] \to \mathbb{R}^d$ solve the uncontrolled SDE*

$$\mathrm{d}X_t = b(X_t, t)\,\mathrm{d}t + \sigma(X_t, t)\,\mathrm{d}B_t, \qquad X_0 = x, \tag{91}$$

where $\sigma(x,t) \in \mathbb{R}^{d \times m}$. Let $f : \mathbb{R}^d \times [0,T] \to \mathbb{R}$, $h : \mathbb{R}^d \times [0,T] \to \mathbb{R}^m$, and $g : \mathbb{R}^d \to \mathbb{R}$ be differentiable in $x$. For $k \in \{1, \ldots, m\}$, denote by $\sigma_{\cdot k}(x,t) \in \mathbb{R}^d$ the $k$-th column of $\sigma(x,t)$, and let

$$G_k(x,t) := \nabla_x \sigma_{\cdot k}(x,t) \in \mathbb{R}^{d \times d}, \qquad \nabla_x h_k(x,t) \in \mathbb{R}^d. \tag{92}$$

Define the adapted process $a : \Omega \times [0,T] \to \mathbb{R}^d$ as the solution of the backward SDE

$$\begin{aligned}
\mathrm{d}a_t = \Big( &- \nabla_x b(X_t,t)^\top a_t - \nabla_x f(X_t,t) + \textstyle\sum_{k=1}^m G_k(X_t,t)^\top \big(G_k(X_t,t)^\top a_t + \nabla_x h_k(X_t,t)\big)\Big)\,\mathrm{d}t \\
&- \textstyle\sum_{k=1}^m \big(G_k(X_t,t)^\top a_t + \nabla_x h_k(X_t,t)\big)\,\mathrm{d}B_t^k, \qquad a_T = \nabla_x g(X_T).
\end{aligned} \tag{93}$$

Then

$$\nabla_{X_0}\big(\textstyle\int_0^T f(X_t,t)\,\mathrm{d}t + \int_0^T \langle h(X_t,t), \mathrm{d}B_t\rangle + g(X_T)\big) = a_0, \tag{94}$$

$$\nabla_x \mathbb{E}\Big[\textstyle\int_0^T f(X_t,t)\,\mathrm{d}t + \int_0^T \langle h(X_t,t), \mathrm{d}B_t\rangle + g(X_T) \,\Big|\, X_0 = x\Big] = \mathbb{E}[a_0], \tag{95}$$

and

$$\begin{aligned}
&\nabla_x \mathbb{E}\Big[\exp\Big(-\textstyle\int_0^T f(X_t,t)\,\mathrm{d}t - \int_0^T \langle h(X_t,t), \mathrm{d}B_t\rangle - g(X_T)\Big) \,\Big|\, X_0 = x\Big] \\
&= -\mathbb{E}\Big[a_0 \exp\Big(-\textstyle\int_0^T f(X_t,t)\,\mathrm{d}t - \int_0^T \langle h(X_t,t), \mathrm{d}B_t\rangle - g(X_T)\Big)\Big].
\end{aligned} \tag{96}$$

Moreover, if $b(x,t) = b_\theta(x,t)$ is differentiable with respect to the parameter $\theta$, and we write $X^\theta$ for the solution,

$$\nabla_\theta\big(\textstyle\int_0^T f(X_t,t)\,\mathrm{d}t + \int_0^T \langle h(X_t,t), \mathrm{d}B_t\rangle + g(X_T)\big) = \int_0^T \big(\nabla_\theta b_\theta(X_t^\theta,t)\big)^\top a_t\,\mathrm{d}t, \tag{97}$$

$$\nabla_\theta \mathbb{E}\Big[\textstyle\int_0^T f(X_t,t)\,\mathrm{d}t + \int_0^T \langle h(X_t,t), \mathrm{d}B_t\rangle + g(X_T) \,\Big|\, X_0 = x\Big] = \mathbb{E}\Big[\int_0^T \big(\nabla_\theta b_\theta(X_t^\theta,t)\big)^\top a_t\,\mathrm{d}t\Big]. \tag{98}$$

**Lemma 9** (Second-order adjoint / Hessian identity for state-dependent diffusion). *Let $X$ solve*

$$\mathrm{d}X_t = b(X_t,t)\,\mathrm{d}t + \sigma(X_t,t)\,\mathrm{d}B_t, \qquad X_0 = x,$$

*with $\sigma(x,t) \in \mathbb{R}^{d \times m}$, and define, for $k \in \{1, \ldots, m\}$,*

$$\sigma_{\cdot k}(x,t) \in \mathbb{R}^d, \qquad G_k(x,t) := \nabla_x \sigma_{\cdot k}(x,t) \in \mathbb{R}^{d \times d}.$$

*Assume $b, \sigma, f, h, g$ are $C^2$ in $x$ (with polynomial growth bounds and moments ensuring the differentiations and Itô steps below are justified).*

*Define the pathwise functional*

$$F(x) := \textstyle\int_0^T f(X_t,t)\,\mathrm{d}t + \int_0^T \langle h(X_t,t), \mathrm{d}B_t\rangle + g(X_T), \qquad X_0 = x, \tag{99}$$

*and its value $J(x) := \mathbb{E}[F(x) \mid X_0 = x]$.*

*Let $(a_t)_{t \in [0,T]}$ be the first-order adjoint from Lemma 8, i.e. the adapted solution of*

$$\begin{aligned}
\mathrm{d}a_t = \Big( &- \nabla_x b(X_t,t)^\top a_t - \nabla_x f(X_t,t) + \textstyle\sum_{k=1}^m G_k(X_t,t)^\top \big(G_k(X_t,t)^\top a_t + \nabla_x h_k(X_t,t)\big)\Big)\,\mathrm{d}t \\
&- \textstyle\sum_{k=1}^m \big(G_k(X_t,t)^\top a_t + \nabla_x h_k(X_t,t)\big)\,\mathrm{d}B_t^k, \qquad a_T = \nabla_x g(X_T).
\end{aligned} \tag{100}$$

*Define also the adapted* matrix-valued *process $A_t \in \mathbb{R}^{d \times d}$ as the solution of the BSDE*

$$\begin{aligned}
\mathrm{d}A_t = -\Big( &\nabla_x b(X_t,t)^\top A_t + A_t \nabla_x b(X_t,t) + \textstyle\sum_{k=1}^m G_k(X_t,t)^\top A_t G_k(X_t,t) + \nabla_x^2 f(X_t,t) \\
&+ \textstyle\sum_{k=1}^m \nabla_x^2 h_k(X_t,t)\, G_k(X_t,t) + \sum_{i=1}^d a_t^{(i)} \nabla_x^2 b_i(X_t,t) + \frac{1}{2}\sum_{k=1}^m \sum_{i=1}^d a_t^{(i)} \nabla_x^2\big(\sigma_{\cdot k}^{(i)}(X_t,t)\big)\, G_k(X_t,t)\Big)\,\mathrm{d}t \\
&+ \textstyle\sum_{k=1}^m U_{t,k}\,\mathrm{d}B_t^k, \qquad A_T = \nabla_x^2 g(X_T),
\end{aligned} \tag{101}$$

where the diffusion matrices $U_{t,k} \in \mathbb{R}^{d \times d}$ are given explicitly by

$$U_{t,k} = -\Big(A_t\, G_k(X_t, t) + G_k(X_t, t)^\top A_t + \nabla_x^2 h_k(X_t, t) + \textstyle\sum_{i=1}^d a_t^{(i)}\, \nabla_x^2 \sigma_{ik}(X_t, t)\Big). \tag{102}$$

Then, for all directions $v, w \in \mathbb{R}^d$, the second directional derivative exists and satisfies the pathwise Hessian-bilinear identity

$$D_{v,w}^2 F(x) \;=\; \langle v,\ A_0\, w \rangle, \tag{103}$$

and consequently,

$$\langle v,\ (\nabla_x^2 J)(x)\, w \rangle \;=\; \mathbb{E}\big[\langle v,\ A_0\, w \rangle\big]. \tag{104}$$

Equivalently, $\nabla_x^2 J(x) = \mathbb{E}[A_0]$ (entrywise) under the same differentiation-under-expectation conditions.

**Lemma 10** (General-$f$ HJB verification step)**.** *Let $u$ be a Markov control and define the cost-to-go*

$$J(u; x, t) = \mathbb{E}\Big[\int_t^T f\big(X_s^u, u(X_s^u, s), s\big)\, \mathrm{d}s + g(X_T^u)\,\Big|\, X_t^u = x\Big], \tag{105}$$

*where $X^u$ solves*

$$\mathrm{d}X_s^u = b\big(X_s^u, u(X_s^u, s), s\big)\, \mathrm{d}s + \sigma\big(X_s^u, u(X_s^u, s), s\big)\, \mathrm{d}B_s, \qquad X_t^u = x. \tag{106}$$

*Assume $J(u; \cdot, \cdot) \in C^{1,2}$ and that, for each $(x, t, p, M)$, the map $\bar{u} \mapsto H(x, t; \bar{u}, p, M)$ attains its minimum, where the Hamiltonian is*

$$H(x, t; \bar{u}, p, M) := f(x, \bar{u}, t) + \langle b(x, \bar{u}, t), p \rangle + \tfrac{1}{2}\mathrm{Tr}\big(\sigma(x, \bar{u}, t)\sigma(x, \bar{u}, t)^\top M\big). \tag{107}$$

*If, for all $(x, t)$,*

$$u(x, t) \in \arg\min_{\bar{u} \in \mathbb{R}^k} H\big(x, t; \bar{u}, \nabla_x J(u; x, t), \nabla_x^2 J(u; x, t)\big), \tag{108}$$

*then $J(u; \cdot, \cdot)$ satisfies the HJB equation*

$$0 = \partial_t J + \inf_{\bar{u} \in \mathbb{R}^k} \Big\{ f(x, \bar{u}, t) + \langle b(x, \bar{u}, t), \nabla_x J \rangle + \tfrac{1}{2}\mathrm{Tr}\big(\sigma(x, \bar{u}, t)\sigma(x, \bar{u}, t)^\top \nabla_x^2 J\big)\Big\}, \qquad J(\cdot, T) = g. \tag{109}$$

*In particular, if the HJB solution is unique, then $J(u; x, t) = V(x, t)$ and $u$ is optimal.*

## B.1 Proof of Lemma 7

Fix a Markov control $u$ and a perturbation $v$, and set $u^\varepsilon := u + \varepsilon v$. Let $X^\varepsilon$ be the controlled diffusion

$$\mathrm{d}X_t^\varepsilon = b\big(X_t^\varepsilon, u^\varepsilon(X_t^\varepsilon, t), t\big)\, \mathrm{d}t + \sigma\big(X_t^\varepsilon, u^\varepsilon(X_t^\varepsilon, t), t\big)\, \mathrm{d}B_t, \qquad X_0^\varepsilon \sim P_0. \tag{110}$$

**Step 1: Value function and martingale identity.** Define the value function associated with the *fixed* control $u$ by

$$J(x, t) := \mathbb{E}\Big[\int_t^T f\big(X_s^{u;t,x}, u(X_s^{u;t,x}, s), s\big)\, \mathrm{d}s + g(X_T^{u;t,x})\Big],$$

where $X^{u;t,x}$ denotes the solution started from $x$ at time $t$ and driven by $u$. Under the stated regularity assumptions, $J \in C^{1,2}$ and satisfies the backward PDE

$$\partial_t J(x, t) + H\big(x, t; u(x, t), \nabla_x J(x, t), \nabla_x^2 J(x, t)\big) = 0, \qquad J(x, T) = g(x),$$

with Hamiltonian

$$H(x, t; u, p, M) = f(x, u, t) + \langle b(x, u, t), p \rangle + \tfrac{1}{2}\mathrm{Tr}\big(\sigma(x, u, t)\sigma(x, u, t)^\top M\big).$$

Apply Itô's formula to the process $t \mapsto J(X_t^\varepsilon, t)$ under the dynamics of $X^\varepsilon$:

$$dJ(X_t^\varepsilon, t) = \Big(\partial_t J + \langle \nabla_x J, b(X_t^\varepsilon, u^\varepsilon(X_t^\varepsilon, t), t)\rangle$$
$$+ \tfrac{1}{2} \operatorname{Tr}\big(\sigma(X_t^\varepsilon, u^\varepsilon(X_t^\varepsilon, t), t)\sigma(X_t^\varepsilon, u^\varepsilon(X_t^\varepsilon, t), t)^\top \nabla_x^2 J\big)\Big) dt$$
$$+ \nabla_x J(X_t^\varepsilon, t)\, \sigma(X_t^\varepsilon, u^\varepsilon(X_t^\varepsilon, t), t)\, dB_t.$$

Adding $f(X_t^\varepsilon, u^\varepsilon(X_t^\varepsilon, t), t)\, dt$ and integrating from 0 to $T$ yields

$$g(X_T^\varepsilon) - J(X_0^\varepsilon, 0) + \int_0^T f(X_t^\varepsilon, u^\varepsilon(X_t^\varepsilon, t), t)\, dt = \int_0^T G_t^\varepsilon\, dt + \int_0^T Z_t^\varepsilon\, dB_t, \tag{111}$$

where

$$G_t^\varepsilon := \partial_t J(X_t^\varepsilon, t) + f(X_t^\varepsilon, u^\varepsilon(X_t^\varepsilon, t), t) + \langle \nabla_x J(X_t^\varepsilon, t), b(X_t^\varepsilon, u^\varepsilon(X_t^\varepsilon, t), t)\rangle$$
$$+ \tfrac{1}{2} \operatorname{Tr}\big(\sigma(X_t^\varepsilon, u^\varepsilon(X_t^\varepsilon, t), t)\sigma(X_t^\varepsilon, u^\varepsilon(X_t^\varepsilon, t), t)^\top \nabla_x^2 J(X_t^\varepsilon, t)\big),$$
$$Z_t^\varepsilon := \nabla_x J(X_t^\varepsilon, t)\, \sigma(X_t^\varepsilon, u^\varepsilon(X_t^\varepsilon, t), t).$$

By the PDE satisfied by $J$, the drift term can be written as the difference of Hamiltonians:

$$G_t^\varepsilon = H\big(X_t^\varepsilon, t;\, u^\varepsilon(X_t^\varepsilon, t), p(X_t^\varepsilon, t), M(X_t^\varepsilon, t)\big) - H\big(X_t^\varepsilon, t;\, u(X_t^\varepsilon, t), p(X_t^\varepsilon, t), M(X_t^\varepsilon, t)\big),$$

where $p = \nabla_x J$ and $M = \nabla_x^2 J$.

Taking expectations in equation (111) and using that the stochastic integral has zero mean, together with the fact that $\mathbb{E}[J(X_0^\varepsilon, 0)]$ does not depend on $\varepsilon$, we obtain

$$\mathcal{J}(u^\varepsilon) - \mathcal{J}(u) = \mathbb{E} \int_0^T \Big(H(X_t^\varepsilon, t; u^\varepsilon(X_t^\varepsilon, t), p(X_t^\varepsilon, t), M(X_t^\varepsilon, t)) - H(X_t^\varepsilon, t; u(X_t^\varepsilon, t), p(X_t^\varepsilon, t), M(X_t^\varepsilon, t))\Big) dt.$$
$$\tag{112}$$

**Step 2: Differentiation at $\varepsilon = 0$.** Divide equation (112) by $\varepsilon$ and let $\varepsilon \to 0$. By the $C^1$ regularity of $H$ in $u$, polynomial growth bounds, and the convergence $X_t^\varepsilon \to X_t^u$ in $L^p$, dominated convergence yields

$$\frac{d}{d\varepsilon}\mathcal{J}(u^\varepsilon)\Big|_{\varepsilon=0} = \mathbb{E} \int_0^T \Big\langle v(X_t^u, t), \nabla_u H\big(X_t^u, t;\, u(X_t^u, t), p_t, M_t\big)\Big\rangle dt,$$

where

$$p_t := \nabla_x J(X_t^u, t), \qquad M_t := \nabla_x^2 J(X_t^u, t).$$

This proves the claim.

### B.2 Proof of Lemma 8

We prove the first identity (94); the identity (95) follows by exchanging the gradient and the expectation, and the exponential identity (96) follows by the chain rule.

**Step 1: directional derivative and tangent process.** Fix a direction $v \in \mathbb{R}^d$ and let

$$\xi_t := \nabla_x X_t\, v \in \mathbb{R}^d \tag{113}$$

denote the directional derivative of $X_t$ with respect to the initial condition. Differentiating equation (91) in the direction $v$ yields the (linear) tangent SDE

$$d\xi_t = \nabla_x b(X_t, t)\, \xi_t\, dt + \sum_{k=1}^m G_k(X_t, t)\, \xi_t\, dB_t^k, \qquad \xi_0 = v. \tag{114}$$

**Step 2: directional derivative of the functional.** Define the pathwise functional

$$F(x) := \int_0^T f(X_t, t) \, dt + \int_0^T \langle h(X_t, t), dB_t \rangle + g(X_T), \qquad X_0 = x. \tag{115}$$

By the chain rule, its directional derivative along $v$ is

$$D_v F(x) = \int_0^T \langle \nabla_x f(X_t, t), \xi_t \rangle \, dt + \sum_{k=1}^m \int_0^T \langle \nabla_x h_k(X_t, t), \xi_t \rangle \, dB_t^k + \langle \nabla_x g(X_T), \xi_T \rangle. \tag{116}$$

**Step 3: choose an adjoint so that an Itô product becomes exact.** Let $a_t$ be an adapted process to be chosen later and consider the scalar product $\langle a_t, \xi_t \rangle$. Apply Itô's product rule using equation (114):

$$d\langle a_t, \xi_t \rangle = \langle da_t, \xi_t \rangle + \langle a_t, d\xi_t \rangle + d\langle a, \xi \rangle_t, \tag{117}$$

where the quadratic covariation term is

$$d\langle a, \xi \rangle_t = \sum_{k=1}^m \left\langle (\text{diffusion coeff. of } a_t \text{ in } dB^k), (\text{diffusion coeff. of } \xi_t \text{ in } dB^k) \right\rangle dt. \tag{118}$$

Write the (unknown) diffusion of $a_t$ as

$$da_t = \alpha_t \, dt + \sum_{k=1}^m \beta_{t,k} \, dB_t^k, \tag{119}$$

with $\alpha_t \in \mathbb{R}^d$, $\beta_{t,k} \in \mathbb{R}^d$ adapted. Then, from equation (114),

$$\langle a_t, d\xi_t \rangle = \langle a_t, \nabla_x b(X_t, t) \xi_t \rangle dt + \sum_{k=1}^m \langle a_t, G_k(X_t, t) \xi_t \rangle dB_t^k, \tag{120}$$

and

$$d\langle a, \xi \rangle_t = \sum_{k=1}^m \langle \beta_{t,k}, G_k(X_t, t) \xi_t \rangle \, dt = \left\langle \sum_{k=1}^m G_k(X_t, t)^\top \beta_{t,k}, \xi_t \right\rangle dt. \tag{121}$$

Substituting into equation (117) gives

$$\begin{aligned} d\langle a_t, \xi_t \rangle &= \left\langle \alpha_t + \nabla_x b(X_t, t)^\top a_t + \sum_{k=1}^m G_k(X_t, t)^\top \beta_{t,k}, \xi_t \right\rangle dt \\ &\quad + \sum_{k=1}^m \left\langle \beta_{t,k} + G_k(X_t, t)^\top a_t, \xi_t \right\rangle dB_t^k. \end{aligned} \tag{122}$$

**Step 4: match the desired $dt$ and $dB_t$ terms.** We would like $d\langle a_t, \xi_t \rangle$ to reproduce (minus) the integrands in equation (116), i.e.

$$d\langle a_t, \xi_t \rangle = -\langle \nabla_x f(X_t, t), \xi_t \rangle \, dt - \sum_{k=1}^m \langle \nabla_x h_k(X_t, t), \xi_t \rangle \, dB_t^k. \tag{123}$$

Comparing equation (122) and equation (123), it suffices to choose $\beta_{t,k}$ and $\alpha_t$ so that, for all $k$,

$$\beta_{t,k} + G_k(X_t, t)^\top a_t = -\nabla_x h_k(X_t, t), \tag{124}$$

$$\alpha_t + \nabla_x b(X_t, t)^\top a_t + \sum_{k=1}^m G_k(X_t, t)^\top \beta_{t,k} = -\nabla_x f(X_t, t). \tag{125}$$

From equation (124), $\beta_{t,k} = -G_k(X_t, t)^\top a_t - \nabla_x h_k(X_t, t)$. Substituting into equation (125) yields

$$\alpha_t = -\nabla_x b(X_t, t)^\top a_t - \nabla_x f(X_t, t) + \sum_{k=1}^m G_k(X_t, t)^\top \big( G_k(X_t, t)^\top a_t + \nabla_x h_k(X_t, t) \big). \tag{126}$$

Therefore the choice equation (93) makes equation (123) hold.

**Step 5: conclude.** Integrating equation (123) from 0 to $T$ gives

$$\langle a_T, \xi_T \rangle - \langle a_0, \xi_0 \rangle = -\int_0^T \langle \nabla_x f(X_t, t), \xi_t \rangle \, dt - \sum_{k=1}^m \int_0^T \langle \nabla_x h_k(X_t, t), \xi_t \rangle \, dB_t^k. \tag{127}$$

Using $\xi_0 = v$ and setting the terminal condition $a_T = \nabla_x g(X_T)$, we obtain

$$\langle a_0, v \rangle = \int_0^T \langle \nabla_x f(X_t, t), \xi_t \rangle \, dt + \sum_{k=1}^m \int_0^T \langle \nabla_x h_k(X_t, t), \xi_t \rangle \, dB_t^k + \langle \nabla_x g(X_T), \xi_T \rangle = D_v F(x), \tag{128}$$

where the last equality is equation (116). Since this holds for all directions $v$, it implies the pathwise identity $\nabla_x F(x) = a_0$.

**Step 6: exchange gradient and expectation.** Under standard regularity and integrability assumptions ensuring that $\nabla_x F(x)$ is integrable and that differentiation under the expectation is valid, taking expectations in equation (146) yields

$$\nabla_x \mathbb{E}\big[F(x) \,\big|\, X_0 = x\big] = \mathbb{E}[\nabla_x F(x)] = \mathbb{E}[a_0], \tag{129}$$

which is the desired identity (95).

Next, we prove the parameter-gradient identities (97) and (98) following an analogous structure. Throughout, fix $\theta$ and write $X_t \equiv X_t^\theta$ for the solution of

$$\mathrm{d}X_t = b_\theta(X_t, t)\,\mathrm{d}t + \sigma(X_t, t)\,\mathrm{d}B_t, \qquad X_0 = x, \tag{130}$$

with $\sigma$ independent of $\theta$. Let $a_t$ be the adjoint process defined in (93) along the trajectory $X^\theta$.

**Step 1': directional derivative in parameter space and parameter tangent process.** Let $\vartheta \in \mathbb{R}^p$ be an arbitrary direction in parameter space and define

$$\eta_t \coloneqq D_\vartheta X_t^\theta \in \mathbb{R}^d, \tag{131}$$

the directional derivative of $X_t^\theta$ with respect to $\theta$ along $\vartheta$. Differentiating equation (130) in direction $\vartheta$ yields the (linear) parameter-tangent SDE

$$\mathrm{d}\eta_t = \Big(\nabla_x b_\theta(X_t, t)\,\eta_t + \nabla_\theta b_\theta(X_t, t)\,\vartheta\Big)\mathrm{d}t + \sum_{k=1}^m G_k(X_t, t)\,\eta_t\,\mathrm{d}B_t^k, \qquad \eta_0 = 0, \tag{132}$$

where $\nabla_\theta b_\theta(x, t) \in \mathbb{R}^{d \times p}$ and $\nabla_\theta b_\theta(x, t)\,\vartheta \in \mathbb{R}^d$. The initial condition is $\eta_0 = 0$ because $X_0 = x$ does not depend on $\theta$.

**Step 2': directional derivative of the functional.** Define the pathwise functional

$$F(\theta) \coloneqq \int_0^T f(X_t, t)\,\mathrm{d}t + \int_0^T \langle h(X_t, t), \mathrm{d}B_t\rangle + g(X_T). \tag{133}$$

By the chain rule, its directional derivative along $\vartheta$ is

$$D_\vartheta F(\theta) = \int_0^T \langle \nabla_x f(X_t, t), \eta_t\rangle\,\mathrm{d}t + \sum_{k=1}^m \int_0^T \langle \nabla_x h_k(X_t, t), \eta_t\rangle\,\mathrm{d}B_t^k + \langle \nabla_x g(X_T), \eta_T\rangle. \tag{134}$$

**Step 3': Itô product with the same adjoint.** Consider the scalar product $\langle a_t, \eta_t\rangle$. Applying Itô's product rule gives

$$\mathrm{d}\langle a_t, \eta_t\rangle = \langle \mathrm{d}a_t, \eta_t\rangle + \langle a_t, \mathrm{d}\eta_t\rangle + \mathrm{d}\langle a, \eta\rangle_t. \tag{135}$$

Write the diffusion of $a_t$ (from (93)) as

$$\mathrm{d}a_t = \alpha_t\,\mathrm{d}t + \sum_{k=1}^m \beta_{t,k}\,\mathrm{d}B_t^k, \tag{136}$$

where

$$\beta_{t,k} = -\big(G_k(X_t, t)^\top a_t + \nabla_x h_k(X_t, t)\big), \tag{137}$$

$$\alpha_t = -\nabla_x b_\theta(X_t, t)^\top a_t - \nabla_x f(X_t, t) + \sum_{k=1}^m G_k(X_t, t)^\top \big(G_k(X_t, t)^\top a_t + \nabla_x h_k(X_t, t)\big). \tag{138}$$

From equation (132),

$$\langle a_t, \mathrm{d}\eta_t\rangle = \Big\langle a_t, \nabla_x b_\theta(X_t, t)\eta_t + \nabla_\theta b_\theta(X_t, t)\,\vartheta\Big\rangle\mathrm{d}t + \sum_{k=1}^m \langle a_t, G_k(X_t, t)\eta_t\rangle\mathrm{d}B_t^k. \tag{139}$$

Moreover, the quadratic covariation term is

$$\mathrm{d}\langle a, \eta\rangle_t = \sum_{k=1}^m \langle \beta_{t,k}, G_k(X_t, t)\eta_t\rangle\,\mathrm{d}t = \Big\langle \sum_{k=1}^m G_k(X_t, t)^\top \beta_{t,k},\, \eta_t\Big\rangle\mathrm{d}t. \tag{140}$$

Substituting equation (136)–equation (140) into equation (135) yields

$$\mathrm{d}\langle a_t, \eta_t\rangle = \Big\langle \alpha_t + \nabla_x b_\theta(X_t, t)^\top a_t + \sum_{k=1}^m G_k(X_t, t)^\top \beta_{t,k},\, \eta_t\Big\rangle\mathrm{d}t + \Big\langle a_t, \nabla_\theta b_\theta(X_t, t)\,\vartheta\Big\rangle\mathrm{d}t$$
$$+ \sum_{k=1}^m \Big\langle \beta_{t,k} + G_k(X_t, t)^\top a_t,\, \eta_t\Big\rangle\mathrm{d}B_t^k. \tag{141}$$

**Step 4': cancel the** $\mathrm{d}t$ **and** $\mathrm{d}B_t$ **terms and keep the forcing term.** By the specific choice equation (93) (equivalently equation (137)), we have for all $k$,

$$\beta_{t,k} + G_k(X_t,t)^\top a_t = -\nabla_x h_k(X_t,t), \qquad \alpha_t + \nabla_x b_\theta(X_t,t)^\top a_t + \sum_{k=1}^m G_k(X_t,t)^\top \beta_{t,k} = -\nabla_x f(X_t,t). \tag{142}$$

Substituting these identities into equation (141) gives

$$\mathrm{d}\langle a_t, \eta_t\rangle = -\langle \nabla_x f(X_t,t), \eta_t\rangle\,\mathrm{d}t - \sum_{k=1}^m \langle \nabla_x h_k(X_t,t), \eta_t\rangle\,\mathrm{d}B_t^k + \Big\langle a_t, \nabla_\theta b_\theta(X_t,t)\,\vartheta\Big\rangle\mathrm{d}t. \tag{143}$$

**Step 5': integrate and conclude the pathwise gradient formula.** Integrating equation (143) from $0$ to $T$ yields

$$\begin{aligned}
\langle a_T, \eta_T\rangle - \langle a_0, \eta_0\rangle = &-\int_0^T \langle \nabla_x f(X_t,t), \eta_t\rangle\,\mathrm{d}t - \sum_{k=1}^m \int_0^T \langle \nabla_x h_k(X_t,t), \eta_t\rangle\,\mathrm{d}B_t^k \\
&+ \int_0^T \Big\langle a_t, \nabla_\theta b_\theta(X_t,t)\,\vartheta\Big\rangle\mathrm{d}t.
\end{aligned} \tag{144}$$

Using $\eta_0 = 0$ and the terminal condition $a_T = \nabla_x g(X_T)$, we rearrange equation (144) to obtain

$$\begin{aligned}
\int_0^T \Big\langle a_t, \nabla_\theta b_\theta(X_t,t)\,\vartheta\Big\rangle\mathrm{d}t &= \int_0^T \langle \nabla_x f(X_t,t), \eta_t\rangle\,\mathrm{d}t + \sum_{k=1}^m \int_0^T \langle \nabla_x h_k(X_t,t), \eta_t\rangle\,\mathrm{d}B_t^k + \langle \nabla_x g(X_T), \eta_T\rangle \\
&= D_\vartheta F(\theta),
\end{aligned} \tag{145}$$

where the last equality is equation (134). Since equation (145) holds for all directions $\vartheta \in \mathbb{R}^p$, it implies the vector identity

$$\nabla_\theta F(\theta) = \int_0^T \big(\nabla_\theta b_\theta(X_t,t)\big)^\top a_t\,\mathrm{d}t. \tag{146}$$

This proves the first desired identity.

**Step 6': exchange gradient and expectation.** Under standard regularity and integrability assumptions ensuring that $\nabla_\theta F(\theta)$ is integrable and that differentiation under the expectation is valid, taking expectations in equation (146) yields

$$\nabla_\theta\,\mathbb{E}\big[F(\theta)\,\big|\,X_0 = x\big] = \mathbb{E}[\nabla_\theta F(\theta)] = \mathbb{E}\Big[\int_0^T \big(\nabla_\theta b_\theta(X_t,t)\big)^\top a_t\,\mathrm{d}t\Big], \tag{147}$$

which is the second desired identity.

### B.3 Proof of Lemma 9

*Proof (sketch).* We prove equation (103); equation (104) follows by exchanging second differentiation and expectation under the assumed integrability bounds.

**Step 1: first and second tangent processes.** Fix $v, w \in \mathbb{R}^d$. Let $\xi_t^v := (\nabla_x X_t)\,v$ and $\xi_t^w := (\nabla_x X_t)\,w$. Differentiating the SDE gives the first-order tangent equations

$$\mathrm{d}\xi_t^\bullet = \nabla_x b(X_t,t)\,\xi_t^\bullet\,\mathrm{d}t + \sum_{k=1}^m G_k(X_t,t)\,\xi_t^\bullet\,\mathrm{d}B_t^k, \qquad \xi_0^\bullet = \bullet, \quad \bullet \in \{v, w\}.$$

Define the mixed second derivative (second tangent) $\zeta_t^{v,w} := D_{v,w}^2 X_t \in \mathbb{R}^d$. A standard differentiation of the tangent SDE yields

$$\begin{aligned}
\mathrm{d}\zeta_t^{v,w} = &\Big(\nabla_x b(X_t,t)\,\zeta_t^{v,w} + \nabla_x^2 b(X_t,t)[\xi_t^v, \xi_t^w]\Big)\,\mathrm{d}t \\
&+ \sum_{k=1}^m \Big(G_k(X_t,t)\,\zeta_t^{v,w} + \nabla_x G_k(X_t,t)[\xi_t^v, \xi_t^w]\Big)\,\mathrm{d}B_t^k, \qquad \zeta_0^{v,w} = 0,
\end{aligned} \tag{148}$$

where $\nabla_x^2 b[\cdot,\cdot] \in \mathbb{R}^d$ denotes the bilinear action of the Hessian tensor of $b$ and $\nabla_x G_k[\cdot,\cdot] \in \mathbb{R}^d$ denotes the bilinear action of the derivative of $G_k$.

**Step 2: second directional derivative of the functional.** Differentiate $F(x)$ twice (chain rule) to obtain

$$
\begin{aligned}
D^2_{v,w} F(x) = \int_0^T & \left( \langle \nabla_x^2 f(X_t,t)\, \xi_t^v, \xi_t^w \rangle + \langle \nabla_x f(X_t,t), \zeta_t^{v,w} \rangle \right) \mathrm{d}t \\
& + \sum_{k=1}^m \int_0^T \left( \langle \nabla_x^2 h_k(X_t,t)\, \xi_t^v, \xi_t^w \rangle + \langle \nabla_x h_k(X_t,t), \zeta_t^{v,w} \rangle \right) \mathrm{d}B_t^k \\
& + \langle \nabla_x^2 g(X_T)\, \xi_T^v, \xi_T^w \rangle + \langle \nabla_x g(X_T), \zeta_T^{v,w} \rangle.
\end{aligned} \tag{149}
$$

**Step 3: eliminate the $\zeta$-terms using the first adjoint.** Apply Itô's product rule to $\langle a_t, \zeta_t^{v,w} \rangle$ using equation (100) and equation (148). Choose the diffusion of $a_t$ as in equation (100) so that the resulting $\mathrm{d}t$ and $\mathrm{d}B_t$ coefficients match $-\langle \nabla_x f, \zeta \rangle$ and $-\langle \nabla_x h_k, \zeta \rangle$. After integrating from 0 to $T$ and using $\zeta_0^{v,w} = 0$ and $a_T = \nabla_x g(X_T)$, one obtains

$$
\begin{aligned}
& \langle \nabla_x g(X_T), \zeta_T^{v,w} \rangle + \int_0^T \langle \nabla_x f(X_t,t), \zeta_t^{v,w} \rangle \, \mathrm{d}t + \sum_{k=1}^m \int_0^T \langle \nabla_x h_k(X_t,t), \zeta_t^{v,w} \rangle \, \mathrm{d}B_t^k \\
& = \int_0^T \left\langle a_t, \nabla_x^2 b(X_t,t)[\xi_t^v, \xi_t^w] \right\rangle \mathrm{d}t + \sum_{k=1}^m \int_0^T \left\langle a_t, \nabla_x G_k(X_t,t)[\xi_t^v, \xi_t^w] \right\rangle \mathrm{d}B_t^k \\
& \qquad\qquad + (\text{terms depending only on } \xi^v, \xi^w),
\end{aligned} \tag{150}
$$

i.e. all $\zeta$-dependence has been traded for terms involving only $(a, \xi^v, \xi^w)$.

**Step 4: matrix adjoint and identification of $U_{t,k}$.** Consider the scalar process $\langle \xi_t^v, A_t \xi_t^w \rangle$ where

$$
\mathrm{d}A_t = \Gamma_t \, \mathrm{d}t + \sum_{k=1}^m U_{t,k} \, \mathrm{d}B_t^k.
$$

Using the first-order tangent SDEs and Itô's formula, the $\mathrm{d}B_t^k$-coefficient of $\mathrm{d}\langle \xi_t^v, A_t \xi_t^w \rangle$ is

$$
\left\langle \xi_t^v, \left( U_{t,k} + A_t G_k(X_t,t) + G_k(X_t,t)^\top A_t \right) \xi_t^w \right\rangle.
$$

To cancel the remaining stochastic integrands in $D^2_{v,w} F(x)$ for all $v, w$, we require

$$
U_{t,k} + A_t G_k + G_k^\top A_t = -\left( \nabla_x^2 h_k + \sum_{i=1}^d a_t^{(i)} \nabla_x^2 \sigma_{ik} \right),
$$

which yields equation (102). With this choice, all $dB$-terms cancel identically.

Matching the $\mathrm{d}t$-terms then uniquely determines the drift $\Gamma_t$, resulting in the BSDE equation (101). Integrating from 0 to $T$, using $\xi_0^v = v$, $\xi_0^w = w$, and $A_T = \nabla_x^2 g(X_T)$, yields

$$
D^2_{v,w} F(x) = \langle v, A_0 w \rangle,
$$

which proves equation (103).

The algebraic details of Steps 3 and 4 (Itô product rule computations and coefficient matching) are standard but lengthy; since the present paper only uses this construction to motivate the BAM objective rather than to develop a new second-order SMP theory, we record the structure here and refer to (Yong & Zhou, 2012, Chapter 3) for the full derivation in the general diffusion setting. $\qquad \square$

### B.4 Proof of Lemma 10

By the dynamic programming principle, for $\Delta t > 0$,

$$
J(u;x,t) = \mathbb{E}[J(u; X_{t+\Delta t}^u, t + \Delta t) \mid X_t^u = x] + \mathbb{E}\left[ \int_t^{t+\Delta t} f(X_s^u, u(X_s^u, s), s) \, \mathrm{d}s \;\middle|\; X_t^u = x \right].
$$

Dividing by $\Delta t$ and sending $\Delta t \to 0$ gives

$$
0 = \mathcal{T}^u J(u;x,t) + f(x, u(x,t), t),
$$

where the controlled generator is

$$\mathcal{T}^u \phi = \partial_t \phi + \langle \nabla_x \phi, \, b(x, u(x,t), t) \rangle + \tfrac{1}{2} \operatorname{Tr}\big( \sigma(x, u(x,t), t) \sigma(x, u(x,t), t)^\top \nabla_x^2 \phi \big).$$

Therefore,

$$0 = \partial_t J + f(x, u(x,t), t) + \langle b(x, u(x,t), t), \nabla_x J \rangle + \tfrac{1}{2} \operatorname{Tr}\big( \sigma(x, u(x,t), t) \sigma(x, u(x,t), t)^\top \nabla_x^2 J \big).$$

If $u(x,t)$ achieves the pointwise minimum in equation (108), then the above identity is equivalent to the HJB equation (109).

## C  Proof of Theorem 2: Lean Adjoint Matching for General Hamiltonians

We follow the same blueprint as (Domingo-Enrich et al., 2025, Prop. 7), but now for a general Hamiltonian and state-dependent diffusion. Let $u$ be an arbitrary control. If $\tilde{a}(t; X^u)$ is the solution of the Lean Adjoint ODE equation (40), it satisfies the integral equation

$$\tilde{a}(t; X^u) = \int_t^T \big( \nabla_1 b(X_s^u, u(X_s^u, s), s)^\top \tilde{a}(s; X^u) + \nabla_1 f(X_s^u, u(X_s^u, s), s) \big)\, ds + \nabla g(X_T^u). \qquad (151)$$

Hence, using the tower property of conditional expectation in the second equality:

$$\mathbb{E}\big[\tilde{a}(t; X^u)\big| X_t^u\big] = \mathbb{E}\Big[ \int_t^T \big( \nabla_1 b(X_s^u, u(X_s^u, s), s)^\top \mathbb{E}\big[\tilde{a}(s; X^u)\big| X_s^u\big] + \nabla_1 f(X_s^u, u(X_s^u, s), s) \big)\, ds + \nabla g(X_T^u) \Big| X_t^u\Big]. \tag{152}$$

Analogously to equations (85), if we consider the perturbation $u^\epsilon(x,t) = u(x,t) + \epsilon v(x,t)$, and let $\bar{u} = \text{stopgrad}(u)$, we have that

$$\begin{aligned}
\frac{\mathrm{d}}{\mathrm{d}\epsilon} \mathbb{E}[\mathcal{L}_{\mathrm{AM}}(u^\epsilon; X^{\bar{u}})]\Big|_{\epsilon=0} &= \mathbb{E}\left[ \int_0^T \Big\langle v(X_t^{\bar{u}}, t), \, \nabla_u \tilde{H}\big(X_t^{\bar{u}}, t; \, \bar{u}(X_t^{\bar{u}}, t), \, \tilde{a}(t, X^{\bar{u}})\big) \Big\rangle \mathrm{d}t \right] \\
&= \mathbb{E}\left[ \int_0^T \Big\langle v(X_t^{\bar{u}}, t), \, \nabla_u \tilde{H}\big(X_t^{\bar{u}}, t; \, \bar{u}(X_t^{\bar{u}}, t), \, \mathbb{E}[\tilde{a}(t, X^{\bar{u}})| X_t^{\bar{u}}]\big) \Big\rangle \mathrm{d}t \right]
\end{aligned} \tag{153}$$

Hence, if $\hat{u}$ is such that the first variation of $u \mapsto \mathbb{E}[\mathcal{L}_{\mathrm{AM}}(u; X^{\bar{u}})]$ at $\hat{u}$ is zero, then

$$\begin{aligned}
0 &= \nabla_u \tilde{H}\big(X_t^{\hat{u}}, t; \, \hat{u}(X_t^{\hat{u}}, t), \, \mathbb{E}[\tilde{a}(t, X^{\hat{u}})| X_t^{\hat{u}}]\big) \\
&= \nabla_2 f(X_t^{\hat{u}}, \hat{u}(X_t^{\hat{u}}, t), t) + \nabla_2 b(X_t^{\hat{u}}, \hat{u}(X_t^{\hat{u}}, t), t)^\top \mathbb{E}[\tilde{a}(t, X^{\hat{u}})| X_t^{\hat{u}}].
\end{aligned} \tag{154}$$

Hence, we have

$$\begin{aligned}
&\nabla_x \hat{u}(X_t^{\hat{u}}, t)^\top \big( \nabla_2 f(X_t^{\hat{u}}, \hat{u}(X_t^{\hat{u}}, t), t) + \nabla_2 b(X_t^{\hat{u}}, \hat{u}(X_t^{\hat{u}}, t), t)^\top \mathbb{E}[\tilde{a}(t, X^{\hat{u}})| X_t^{\hat{u}}]\big) = 0, \\
&\implies \mathbb{E}\big[ \int_t^T \nabla_x \hat{u}(X_s^{\hat{u}}, s)^\top \big( \nabla_2 f(X_s^{\hat{u}}, \hat{u}(X_s^{\hat{u}}, s), s) + \nabla_2 b(X_s^{\hat{u}}, \hat{u}(X_s^{\hat{u}}, s), s)^\top \mathbb{E}[\tilde{a}(s, X^{\hat{u}})| X_s^{\hat{u}}]\big)\, \mathrm{d}s \big| X_t^{\hat{u}}\big] = 0.
\end{aligned} \tag{155}$$

If we add equation (155) to the right-hand side of equation (152), we obtain that $\mathbb{E}[\tilde{a}(t, X^{\hat{u}})| X_t^{\hat{u}}]$ also solves the following integral equation:

$$\begin{aligned}
&\mathbb{E}\big[\tilde{a}(t; X^{\hat{u}})\big| X_t^{\hat{u}}\big] \\
&= \mathbb{E}\Big[ \int_t^T \big( (\nabla_1 b(X_s^{\hat{u}}, \hat{u}(X_s^{\hat{u}}, s), s) + \nabla_2 b(X_s^{\hat{u}}, \hat{u}(X_s^{\hat{u}}, s), s) \nabla_x \hat{u}(X_s^{\hat{u}}, s))^\top \mathbb{E}\big[\tilde{a}(s; X^{\hat{u}})\big| X_s^{\hat{u}}\big] \\
&\qquad\quad + \nabla_1 f(X_s^{\hat{u}}, \hat{u}(X_s^{\hat{u}}, s), s) + \nabla_x \hat{u}(X_s^{\hat{u}}, s)^\top \nabla_2 f(X_s^{\hat{u}}, \hat{u}(X_s^{\hat{u}}, s), s) \big)\, \mathrm{d}s + \nabla g(X_T^{\hat{u}}) \Big| X_t^{\hat{u}}\Big] \\
&= \mathbb{E}\Big[ \int_t^T \big( \nabla_x b(x, \hat{u}(x,s), s)|_{x=X_s^{\hat{u}}}^\top \mathbb{E}\big[\tilde{a}(s; X^{\hat{u}})\big| X_s^{\hat{u}}\big] + \nabla_x f(x, \hat{u}(x,s), s)|_{x=X_s^{\hat{u}}} \big)\, \mathrm{d}s + \nabla g(X_T^{\hat{u}}) \Big| X_t^{\hat{u}}\Big].
\end{aligned} \tag{156}$$

Now, observe that when $\sigma(x, u, t) := \sigma(t)$, the adjoint SDE (30)-(31) simplifies to the lean adjoint ODE:

$$\frac{\mathrm{d}a_t}{\mathrm{d}t} = -\nabla_x b(x, u(x,t), t)\big|_{x=X_t}^\top a_t - \nabla_x f(x, u(x,t), t)\big|_{x=X_t}, \tag{157}$$

$$a_T = \nabla_x g(X_T), \tag{158}$$

which, setting $u = \hat{u}$, means that $\mathbb{E}\big[a(t; X^{\hat{u}})\big|X_t^{\hat{u}}\big]$ satisfies the following integral equation:

$$
\begin{aligned}
&\mathbb{E}\big[a(t; X^{\hat{u}})\big|X_t^{\hat{u}}\big] \\
&= \mathbb{E}\Big[ \int_t^T \big(\nabla_x b(x, \hat{u}(x, s), s)\big|_{x=X_s^{\hat{u}}}^\top \, \mathbb{E}\big[a(s; X^{\hat{u}})\big|X_s^{\hat{u}}\big] + \nabla_x f(x, \hat{u}(x, s), s)\big|_{x=X_s^{\hat{u}}}\big)\,\mathrm{d}s + \nabla g(X_T^{\hat{u}})\Big|X_t^{\hat{u}}\Big],
\end{aligned}
\tag{159}
$$

Comparing equation (156) with equation (159), we see that $\mathbb{E}\big[\tilde{a}(t; X^{\hat{u}})\big|X_t^{\hat{u}}\big]$ and $\mathbb{E}\big[a(t; X^{\hat{u}})\big|X_t^{\hat{u}}\big]$ satisfy the same integral equation. Applying Lemma 11, we obtain that

$$
\mathbb{E}\big[\tilde{a}(t; X^{\hat{u}})\big|X_t^{\hat{u}}\big] = \mathbb{E}\big[a(t; X^{\hat{u}})\big|X_t^{\hat{u}}\big].
\tag{160}
$$

We now evaluate both the BAM and AM first variations at the critical point $u = \hat{u}$ (so that $\bar{u} = \hat{u}$ and all trajectory superscripts coincide). By equations (85) and (86),

$$
\begin{aligned}
&\frac{\mathrm{d}}{\mathrm{d}\epsilon} \mathbb{E}[\mathcal{L}_{\mathrm{BAM}}(\hat{u} + \epsilon v; X^{\hat{u}})]\Big|_{\epsilon=0} \\
&= \mathbb{E}\Big[ \int_0^T \Big\langle v(X_t^{\hat{u}}, t),\, \nabla_u H\big(X_t^{\hat{u}}, t;\, \hat{u}(X_t^{\hat{u}}, t),\, \mathbb{E}\big[a(t; X^{\hat{u}})\big|X_t^{\hat{u}}\big],\, \mathbb{E}\big[A(t; X^{\hat{u}})\big|X_t^{\hat{u}}\big]\big)\Big\rangle \,\mathrm{d}t\Big] \\
&= \mathbb{E}\Big[ \int_0^T \Big\langle v(X_t^{\hat{u}}, t),\, \nabla_u H\big(X_t^{\hat{u}}, t;\, \hat{u}(X_t^{\hat{u}}, t),\, \mathbb{E}\big[\tilde{a}(t; X^{\hat{u}})\big|X_t^{\hat{u}}\big]\big)\Big\rangle \,\mathrm{d}t\Big] = \frac{\mathrm{d}}{\mathrm{d}\epsilon} \mathbb{E}[\mathcal{L}_{\mathrm{AM}}(\hat{u} + \epsilon v; X^{\hat{u}})]\Big|_{\epsilon=0}.
\end{aligned}
\tag{161}
$$

Here, the second equality uses that since $\sigma(x, u, t) = \sigma(t)$, the second-order adjoint term drops out of $\nabla_u H$:

$$
\begin{aligned}
&\nabla_u H\big(X_t^{\hat{u}}, t;\, \hat{u}(X_t^{\hat{u}}, t),\, \mathbb{E}\big[a(t; X^{\hat{u}})\big|X_t^{\hat{u}}\big],\, \mathbb{E}\big[A(t; X^{\hat{u}})\big|X_t^{\hat{u}}\big]\big) = \nabla_u H\big(X_t^{\hat{u}}, t;\, \hat{u}(X_t^{\hat{u}}, t),\, \mathbb{E}\big[a(t; X^{\hat{u}})\big|X_t^{\hat{u}}\big]\big) \\
&= \nabla_u H\big(X_t^{\hat{u}}, t;\, \hat{u}(X_t^{\hat{u}}, t),\, \mathbb{E}\big[\tilde{a}(t; X^{\hat{u}})\big|X_t^{\hat{u}}\big]\big),
\end{aligned}
\tag{162}
$$

where the last step uses equation (160). The rest of the proof is the same as the end of the proof of Theorem 1.

**Lemma 11.** *Let $X^u$ be a solution of the SDE (4), and let $\mathcal{A} : \mathbb{R}^d \times [0, T] \to \mathbb{R}^{d \times d}$ satisfy $\sup_{t \in [0,T]} \mathbb{E}[\|\mathcal{A}(X_t^u, t)\|^2] < +\infty$. Consider the integral equation*

$$
Y_t = \mathbb{E}\Big[ \int_t^T \big(\mathcal{A}(X_s^u, s)Y_s + c(X_s^u, s)\big)\mathrm{d}s + \psi(X_T^u)\Big|X_t^u\Big],
\tag{163}
$$

*where $c : \mathbb{R}^d \times [0, T] \to \mathbb{R}^d$ and $\psi : \mathbb{R}^d \to \mathbb{R}^d$ are measurable with appropriate integrability, and $Y_t$ is understood as a function of $X_t^u$. Then this equation has a unique solution.*

*Proof.* Let $Y^1, Y^2$ be two solutions. Their difference satisfies

$$
Y_t^1 - Y_t^2 = \mathbb{E}\Big[ \int_t^T \mathcal{A}(X_s^u, s)\big(Y_s^1 - Y_s^2\big)\mathrm{d}s\Big|X_t^u\Big].
\tag{164}
$$

Define $\phi(t) := \mathbb{E}\big[\|Y_t^1 - Y_t^2\|\big]$. Taking norms inside the conditional expectation and then full expectations:

$$
\begin{aligned}
\phi(t) &\leq \mathbb{E}\Big[ \int_t^T \|\mathcal{A}(X_s^u, s)\| \cdot \|Y_s^1 - Y_s^2\| \,\mathrm{d}s\Big] = \int_t^T \mathbb{E}\big[\|\mathcal{A}(X_s^u, s)\| \cdot \|Y_s^1 - Y_s^2\|\big] \,\mathrm{d}s \\
&\leq \int_t^T \big(\mathbb{E}\big[\|\mathcal{A}(X_s^u, s)\|^2\big]\big)^{1/2} \phi(s) \,\mathrm{d}s \leq C \int_t^T \phi(s) \,\mathrm{d}s,
\end{aligned}
\tag{165}
$$

where $C := \sup_{s \in [0,T]} \big(\mathbb{E}[\|\mathcal{A}(X_s^u, s)\|^2]\big)^{1/2} < +\infty$. By the backward Grönwall inequality, $\phi(t) = 0$ for all $t \in [0, T]$. Hence $Y_t^1 = Y_t^2$ a.s. for every $t$. $\qquad\square$

# D   Stochastic Maximum Principle

## D.1   Value function representation of the SMP adjoint processes

We first examine the SMP adjoint processes (22)–(25) when the diffusion is control-independent, and prove the identities stated in equation (26)

$$
p_t^* = \nabla_x V(X_t^*, t), \qquad q_t^* = \nabla_x^2 V(X_t^*, t)\,\sigma(X_t^*, t),
$$

where $V(x,t) := J(u^*; x, t)$ is the value function under the optimal control $u^*$. We assume $V$ is sufficiently smooth so that the computations below are justified. The argument proceeds in three steps: (1) apply Itô's formula to $\nabla_x V(X_t^*, t)$ to identify the diffusion coefficient, (2) use the HJB equation to simplify the drift, and (3) invoke BSDE uniqueness.

Throughout, we write $b^* := b(X_t^*, u_t^*, t)$, $\sigma^* := \sigma(X_t^*, t)$, and $\Sigma^* := \sigma^*(\sigma^*)^\top$ for brevity.

**Step 1: Itô's formula.** Define $P_t := \nabla_x V(X_t^*, t) \in \mathbb{R}^d$. Applying Itô's formula componentwise to $\partial_{x_i} V(X_t^*, t)$ gives

$$\mathrm{d}P_t = \left[\partial_t \nabla_x V + \nabla_x^2 V\, b^* + \tfrac{1}{2}\,\mathcal{T}\right]\mathrm{d}t + \nabla_x^2 V\, \sigma^*\, \mathrm{d}B_t, \tag{166}$$

where $\mathcal{T} \in \mathbb{R}^d$ has components $\mathcal{T}_i := \sum_{k,\ell} \partial_{x_k x_\ell x_i} V(X_t^*, t)\, \Sigma_{k\ell}^*$, and all functions are evaluated at $(X_t^*, t)$. Setting $Q_t := \nabla_x^2 V(X_t^*, t)\, \sigma(X_t^*, t)$, we read off the diffusion coefficient directly:

$$Q_t = \nabla_x^2 V(X_t^*, t)\, \sigma(X_t^*, t). \tag{167}$$

**Step 2: Simplifying the drift via the HJB equation.** The value function satisfies the HJB equation

$$0 = \partial_t V + \inf_u \left\{ f(x,u,t) + \langle \nabla_x V,\, b(x,u,t)\rangle + \tfrac{1}{2}\operatorname{Tr}\bigl(\Sigma(x,t)\,\nabla_x^2 V\bigr)\right\}, \tag{168}$$

where $\Sigma(x,t) := \sigma(x,t)\sigma(x,t)^\top$. Evaluating at the optimal control $u^*$ and differentiating with respect to $x$, we obtain

$$0 = \partial_t \nabla_x V + \nabla_1 f^* + (\nabla_1 b^*)^\top \nabla_x V + \nabla_x^2 V\, b^* + \tfrac{1}{2}\nabla_x \operatorname{Tr}\bigl(\Sigma^* \nabla_x^2 V\bigr), \tag{169}$$

where $\nabla_1 f^* := \nabla_1 f(X_t^*, u_t^*, t)$ and $\nabla_1 b^* := \nabla_1 b(X_t^*, u_t^*, t)$ denote partial derivatives with respect to the first argument $x$. Expanding the last term componentwise:

$$\left[\nabla_x \operatorname{Tr}\bigl(\Sigma \nabla_x^2 V\bigr)\right]_i = \sum_{k,\ell}(\partial_{x_i}\Sigma_{k\ell})\,\partial_{x_k x_\ell} V + \underbrace{\sum_{k,\ell}\Sigma_{k\ell}\,\partial_{x_k x_\ell x_i} V}_{\mathcal{T}_i}. \tag{170}$$

Solving equation (169) for $\partial_t \nabla_x V$ and substituting into the Itô drift equation (166), the $\nabla_x^2 V\, b^*$ terms cancel and the $\mathcal{T}$ terms cancel, leaving

$$\mathrm{d}P_t = -\left[\nabla_1 f^* + (\nabla_1 b^*)^\top P_t + \tfrac{1}{2}\sum_{k,\ell}(\nabla_x \Sigma_{k\ell}^*)\,\partial_{x_k x_\ell} V\right]\mathrm{d}t + Q_t\, \mathrm{d}B_t. \tag{171}$$

**Step 3: Matching with the SMP adjoint BSDE.** It remains to show that the drift in equation (171) equals $-\nabla_x \mathcal{H}(X_t^*, t; u_t^*, P_t, Q_t)$, where

$$\nabla_x \mathcal{H}(x, t; u, p, q) = \nabla_1 f(x,u,t) + \nabla_1 b(x,u,t)^\top p + \nabla_x \operatorname{Tr}\bigl(\sigma(x,t)^\top q\bigr), \tag{172}$$

and the last term differentiates $\sigma(x,t)$ with respect to $x$ while holding $q$ fixed. Evaluating at $q = Q_t = \nabla_x^2 V\,\sigma$:

$$\left[\nabla_x \operatorname{Tr}\bigl(\sigma^\top Q_t\bigr)\right]_i = \sum_{j,k}(\partial_{x_i}\sigma_{jk})\,(Q_t)_{jk} = \sum_{j,k}(\partial_{x_i}\sigma_{jk})\sum_\ell (\partial_{x_j x_\ell} V)\,\sigma_{\ell k}. \tag{173}$$

Using the identity $\partial_{x_i}\Sigma_{j\ell} = \sum_k[(\partial_{x_i}\sigma_{jk})\,\sigma_{\ell k} + \sigma_{jk}\,(\partial_{x_i}\sigma_{\ell k})]$ together with the symmetry $\partial_{x_j x_\ell} V = \partial_{x_\ell x_j} V$, one verifies that

$$\nabla_x \operatorname{Tr}\bigl(\sigma^\top Q_t\bigr) = \tfrac{1}{2}\sum_{k,\ell}(\nabla_x \Sigma_{k\ell})\,\partial_{x_k x_\ell} V. \tag{174}$$

Substituting equation (174) into equation (171) shows that $P_t$ satisfies the BSDE

$$\mathrm{d}P_t = -\nabla_x \mathcal{H}\bigl(X_t^*, t; u_t^*, P_t, Q_t\bigr)\, \mathrm{d}t + Q_t\, \mathrm{d}B_t, \qquad P_T = \nabla_x g(X_T^*), \tag{175}$$

where the terminal condition follows from $V(x,T) = g(x)$. This is exactly the SMP adjoint BSDE equation (23). By uniqueness of the adapted solution, we conclude $p_t^* = P_t = \nabla_x V(X_t^*, t)$ and $q_t^* = Q_t = \nabla_x^2 V(X_t^*, t)\, \sigma(X_t^*, t)$.

## D.2 Second-order statement for control-dependent diffusions

To state the stochastic maximum principle (SMP) in full generality, we first recall the *first-order Hamiltonian*

$$\mathcal{H}(x,t;u,p,q) = f(x,u,t) + \langle p, b(x,u,t) \rangle + \mathrm{Tr}\big(\sigma(x,u,t)^\top q\big), \text{ for } (x,t,u,p,q) \in \mathbb{R}^d \times [0,T] \times U \times \mathbb{R}^d \times \mathbb{R}^{d \times m}. \tag{176}$$

In this case, $U \subseteq \mathbb{R}^k$ is the control domain. Let $u_t^* := u^*(X_t^*, t)$ be an optimal control and $X_t^*$ its associated state process. The *first-order adjoint processes* $(p_t^*, q_t^*)$ are defined as the solution of the backward stochastic differential equation

$$\begin{cases} \mathrm{d}p_t^* = -\nabla_x \mathcal{H}\big(X_t^*, t; u_t^*, p_t^*, q_t^*\big)\,\mathrm{d}t + q_t^*\,\mathrm{d}B_t, & (177) \\ p_T^* = \nabla_x g(X_T^*). & (178) \end{cases}$$

This backward equation remains the same as equation (23) in Section 2.4 when $\sigma$ does not depend on the control $u$.

**Second-order adjoint.** When the control domain $U$ is nonconvex *and* the diffusion coefficient depends on the control, the first-order adjoint alone is not sufficient to characterize optimality. In this case, the SMP introduces a *second-order adjoint* $(P_t^*, Q_t^*)$, where $P_t^* \in \mathbb{S}^d$ is symmetric and $Q_t^* = (Q_t^{*,1}, \dots, Q_t^{*,m})$ with $Q_t^{*,k} \in \mathbb{R}^{d \times d}$. The second-order adjoint satisfies the matrix-valued BSDE

$$\begin{aligned} \mathrm{d}P_t^* = -\Big( & \nabla_1 b(X_t^*, u(X_t^*, t), t)^\top P_t^* + P_t^* \nabla_1 b(X_t^*, u(X_t^*, t), t) \\ & + \sum_{k=1}^m \nabla_1 \sigma_{\cdot k}(X_t^*, u(X_t^*, t), t)^\top P_t^* \nabla_1 \sigma_{\cdot k}(X_t^*, u(X_t^*, t), t) \\ & + \sum_{k=1}^m \big(\nabla_1 \sigma_{\cdot k}(X_t^*, u(X_t^*, t), t)^\top Q^{*,k} + Q^{*,k} \nabla_1 \sigma_{\cdot k}(X_t^*, u(X_t^*, t), t)\big) \\ & + \nabla_x^2 \mathcal{H}(X_t^*, t; u_t^*, p_t^*, q_t^*)\Big)\mathrm{d}t \end{aligned} \tag{179}$$

$$P_T^* = \nabla_x^2 g(X_T^*). \tag{180}$$

where all derivatives are evaluated along $(X_t^*, u_t^*)$.

**Generalized Hamiltonian and optimality condition.** Using the second-order adjoint, define the *generalized Hamiltonian*

$$\mathcal{H}_{\mathrm{SMP}}(x,t;u) = \mathcal{H}(x,t;u,p_t^*,q_t^*) + \tfrac{1}{2}\mathrm{Tr}\Big(\big(\sigma(x,u,t) - \sigma(x,u_t^*,t)\big)^\top P_t^*\big(\sigma(x,u,t) - \sigma(x,u_t^*,t)\big)\Big). \tag{181}$$

The stochastic maximum principle (Yong & Zhou, 2012, Theorem 3.2) states that, for almost every $t$ and almost surely with respect to $X^*$,

$$\mathcal{H}_{\mathrm{SMP}}(X_t^*, t; u_t^*) = \min_{u \in \mathbb{R}^k} \mathcal{H}_{\mathrm{SMP}}(X_t^*, t; u). \tag{182}$$

**First-order SMP as a special case.** If the diffusion coefficient $\sigma$ does not depend on the control, *or* if the control domain $U \subseteq \mathbb{R}^d$ is convex so that second-order variations vanish[3], the quadratic correction term in equation (181) disappears. In this case, the SMP reduces to the first-order condition

$$\mathcal{H}\big(X_t^*, t; u_t^*, p_t^*, q_t^*\big) = \min_{u \in \mathbb{R}^k} \mathcal{H}\big(X_t^*, t; u, p_t^*, q_t^*\big), \tag{183}$$

together with the state equation and the first-order adjoint BSDE equation (177). This is the setting considered in the main text of the paper.

---

[3]When the control domain $U$ is convex, admissible perturbations of an optimal control can be taken as convex combinations rather than spike variations. In this case, even if the diffusion coefficient depends on the control, first-order variations suffice and the stochastic maximum principle yields a first-order variational inequality $\langle \nabla_u \mathcal{H}(X_t^*, t; u_t^*, p_t^*, q_t^*), u - u_t^* \rangle \geq 0$ for all $u \in U$, which is equivalent (under standard convexity assumptions on $\mathcal{H}$ in $u$) to pointwise minimization of the Hamiltonian over $U$. By contrast, when $U$ is nonconvex and the diffusion depends on the control, spike variations produce $O(\varepsilon)$ quadratic diffusion terms, necessitating a second-order adjoint and a generalized Hamiltonian.

# E   Proof of Theorem 5

To show that $\frac{\mathrm{d}}{\mathrm{d}\tau}u^\tau = -\frac{\mathrm{d}}{\mathrm{d}u}\mathbb{E}\big[\int_0^T \tilde{H}(X_t^\tau, t; u(X_t^\tau, t), p_t^\tau)\,\mathrm{d}t\big]\big|_{u=u^\tau}$ and $\frac{\mathrm{d}}{\mathrm{d}\tau}u^\tau = -\frac{\mathrm{d}}{\mathrm{d}u}\mathbb{E}\big[\mathcal{L}_{\mathrm{AM}}(u; X^{\bar{u}})\big]\big|_{u=u^\tau}$ are the same equation, it suffices to show that $\mathbb{E}\big[\int_0^T \tilde{H}(X_t^\tau, t; u(X_t^\tau, t), p_t^\tau)\,\mathrm{d}t\big] = \mathbb{E}\big[\mathcal{L}_{\mathrm{AM}}(u; X^{u^\tau})\big]$ for all functions $u : \mathbb{R}^d \times [0, T] \to \mathbb{R}$. By the tower property, we have that

$$\mathbb{E}\big[\int_0^T \tilde{H}(X_t^\tau, t; u(X_t^\tau, t), p_t^\tau)\,\mathrm{d}t\big] = \mathbb{E}\big[\int_0^T \tilde{H}(X_t^\tau, t; u(X_t^\tau, t), \mathbb{E}[p_t^\tau | X_t^\tau])\,\mathrm{d}t\big] \tag{184}$$

And by the definition of $\mathcal{L}_{\mathrm{AM}}(u; X^{u^\tau})$, the identity $X^\tau = X^{u^\tau}$, and the tower property, we obtain the following:

$$\begin{aligned}
\mathbb{E}\big[\mathcal{L}_{\mathrm{AM}}(u; X^{u^\tau})\big] &= \mathbb{E}\big[\int_0^T \tilde{H}\big(X_t^{u^\tau}, t; u(X_t^{u^\tau}, t), \tilde{a}(t; X^{u^\tau})\big)\,\mathrm{d}t\big] \\
&= \mathbb{E}\big[\int_0^T \tilde{H}\big(X_t^\tau, t; u(X_t^\tau, t), \tilde{a}(t; X^\tau)\big)\,\mathrm{d}t\big] = \mathbb{E}\big[\int_0^T \tilde{H}\big(X_t^\tau, t; u(X_t^\tau, t), \mathbb{E}[\tilde{a}(t; X^\tau) | X_t^\tau]\big)\,\mathrm{d}t\big]
\end{aligned} \tag{185}$$

Hence, it suffices to show the following equality:

$$\mathbb{E}[p_t^\tau | X_t^\tau] = \mathbb{E}[\tilde{a}(t; X^\tau) | X_t^\tau]. \tag{186}$$

Using the argument in equation (152), $\mathbb{E}\big[\tilde{a}(t; X^\tau)\big| X_t^\tau\big]$ satisfies the integral equation

$$\mathbb{E}\big[\tilde{a}(t; X^\tau)\big| X_t^\tau\big] = \mathbb{E}\Big[\int_t^T \big(\nabla_1 b(X_s^\tau, u^\tau(X_s^\tau, s), s)^\top \mathbb{E}\big[\tilde{a}(s; X^\tau)\big| X_s^\tau\big] + \nabla_1 f(X_s^\tau, u^\tau(X_s^\tau, s), s)\big)\,\mathrm{d}s + \nabla g(X_T^\tau)\Big| X_t^\tau\Big]. \tag{187}$$

Similarly, we can write an integral equation for the evolution of $p_t^\tau$:

$$p_t^\tau = \int_t^T \big((p_s^\tau)^\top \nabla_1 b(X_s^\tau, u^\tau(X_s^\tau, s), s) + \nabla_1 f(X_s^\tau, u^\tau(X_s^\tau, s), s)\big)\mathrm{d}s - \int_t^T q_s^\tau \mathrm{d}B_s + \nabla g(X_T^\tau). \tag{188}$$

Taking conditional expectations of the integral form of the BSDE defining $p_t^\tau$ with respect to $X_t^\tau$, and using the tower property together with the martingale property of the stochastic integral, we obtain

$$\mathbb{E}[p_t^\tau | X_t^\tau] = \mathbb{E}\Big[\int_t^T \Big(\mathbb{E}[p_s^\tau | X_s^\tau]^\top \nabla_1 b(X_s^\tau, u^\tau(X_s^\tau, s), s) + \nabla_1 f(X_s^\tau, u^\tau(X_s^\tau, s), s)\Big)\,\mathrm{d}s + \nabla g(X_T^\tau)\Big| X_t^\tau\Big]. \tag{189}$$

That is, $\mathbb{E}[p_t^\tau | X_t^\tau]$ satisfies the same state-conditioned integral equation above as the lean adjoint conditional expectation $\mathbb{E}[\tilde{a}(t; X^\tau) | X_t^\tau]$ in equation (187) with $u = u^\tau$. Thus, by Lemma 11 we obtain that

$$\mathbb{E}[p_t^\tau | X_t^\tau] = \mathbb{E}[\tilde{a}(t; X^\tau) | X_t^\tau], \tag{190}$$

which concludes the proof.

## E.1   Second proof approach in the path integral setting: using the Feynman-Kac formula

In path integral SOC, or quadratic-cost control-affine SOC, we have that $b(x, u, t) = \bar{b}(x, t) + \sigma(x, t)u$ and $f(x, u, t) = \bar{f}(x, t) + \frac{1}{2}\|u(x, t)\|^2$. We give another explanation to equation (186) from the viewpoint of the Feynman-Kac formula (Theorem 12). For the backward stochastic equation for $p_t^\tau$, the solution $p_t^\tau$ can be written as

$$p_t^\tau = \mathbb{E}\Big[\Phi_{t,T}\,\nabla_x g(X_T^\tau) + \int_t^T \Phi_{t,s}\,\nabla_x \bar{f}(X_s^\tau, s)\mathrm{d}s \;\Big|\; \mathcal{F}_t\Big], \tag{191}$$

where $\Phi_{t,s}$ is the fundamental matrix of $\mathrm{d}\Phi_{t,s}/\mathrm{d}s = \nabla_x \bar{b}(X_s^\tau, s)\,\Phi_{t,s}$, $\Phi_{t,t} = I$.

Therefore, given a trajectory $\{X_t^\tau\}_{t=0}^T$ satisfying (49),

$$\Phi_{t,T}\,\nabla_x g(X_T^\tau) + \int_t^T \Phi_{t,s}\,\nabla_x \bar{f}(X_s^\tau, s)\mathrm{d}s$$

is an *unbiased estimator* of $\mathbb{E}[p_t^\tau | X_t^\tau]$. Meanwhile, it is straightforward to verify that

$$\tilde{a}(t; X^\tau) = \Phi_{t,T} \nabla_x g(X_T^\tau) + \int_t^T \Phi_{t,s} \nabla_x \bar{f}(X_s^\tau, s) \mathrm{d}s. \tag{192}$$

Therefore we have

$$\mathbb{E}[p_t^\tau | X_t^\tau] = \mathbb{E}[\tilde{a}(t; X^\tau) | X_t^\tau]. \tag{193}$$

From this perspective, having a good surrogate for the diffusion term $q_t^\tau$ may lead to a better unbiased estimator with reduced variance.

**Theorem 12** (Feynman–Kac representation for a linear BSDE). *Let $(X_t)_{t \in [0,T]}$ solve the forward SDE*

$$\mathrm{d}X_t = b(X_t, t) \, \mathrm{d}t + \sigma(X_t, t) \, \mathrm{d}B_t. \tag{194}$$

*Assume that $A(\cdot, \cdot)$ and $c(\cdot, \cdot)$ are progressively measurable and bounded, and that $\psi(X_T) \in L^2(\Omega)$. Then there exists a unique adapted solution $(Y, Z) \in \mathcal{S}^2 \times \mathcal{H}^2$ to the linear BSDE*

$$\begin{cases} \mathrm{d}Y_t = -\big(A(X_t, t)^\top Y_t + c(X_t, t)\big) \, \mathrm{d}t + Z_t \, \mathrm{d}B_t, \\ Y_T = \psi(X_T). \end{cases} \tag{195}$$

*Moreover, the process $Y_t$ admits the representation*

$$Y_t = \mathbb{E}\Big[ \Phi_{t,T} \, \psi(X_T) + \int_t^T \Phi_{t,s} \, c(X_s, s) \, \mathrm{d}s \,\Big|\, \mathcal{F}_t \Big], \tag{196}$$

*where $\Phi_{t,s}$ is the* fundamental matrix *(state-transition matrix) solving*

$$\tfrac{\mathrm{d}}{\mathrm{d}s} \Phi_{t,s} = A(X_s, s) \, \Phi_{t,s}, \qquad \Phi_{t,t} = I. \tag{197}$$

*When $d = 1$, this reduces to the scalar exponential $\Phi_{t,s} = \exp\big( \int_t^s A(X_\tau, \tau) \, \mathrm{d}\tau \big)$. The process $Z_t$ is uniquely determined by $Y_t$ through the martingale representation theorem.*

