# OpenReview forum: "Adjoint Matching through the Lens of the Stochastic Maximum Principle in Optimal Control"
_TMLR — Decision pending for TMLR_

### Review · Reviewer_NXPS · 2026-05-16

**Summary Of Contributions:**

## Summary

This paper studies the adjoint matching framework, recently introduced
to design fine-tuning methods for diffusion models; particularly, this
work derives adjoint-matching-type optimization schemes from the
bottom-up as opposed to top-down like in the seminal work. As a
consequence, the bottom-up approach admits optimization schemes for a
broader family of diffusion dynamics coefficients, and recovers existing
methods as a special case. Furthermore, the paper establishes that this
optimization framework can be interpreted as a clever proxy for the
method of successive approximations (MSA) in the stochastic case, which
may suggest algorithms for stochastic control problems based on
Pontryagin's Maximum Principle beyond diffusion model finetuning.

## Strengths

Besides some more minor and superficial writing criticims that I mention
later, the text is largely well written, and the mathematical precision
and notation are adequate. As far as I can tell, the mathematical claims
are correct. It is really nice to see classical control theory revived
in this work, and the authors do a nice job of contextualizing it in the
more modern diffusion model setting.

## Weaknesses

The main criticism I have is that the actual contributions of the paper
are not obvious. The paper is entirely theoretical, which is fine,
however it also does not really provide any new algorithms, and does not
comment much on existing algorithms either. The exception is the
algorithm of Domingo-Enrich, which the authors identify as a particular
instance of the equation derived in Theorem 2, but there is no
commentary on what Theorem 2 provides in addition. Similarly, while
section 5 appears to draw a deep connection between existing adjoint
matching methods and the stochastic maximum principle, there is again no
follow-up: as a reader, I do not understand how this connection
influences either adjoint matching algorithms or algorithms based on the
stochastic maximum principle.

**Audience:**

No

**Audience Explanation:**

It feels harsh to write "no" to such a question, but as mentioned in
**Weaknesses**, I believe the paper does not provide enough actionable
information to a machine learning audience – it is not clear to me what
influence these results actually have on fine-tuning algorithms for
diffusion models, or even stochastic control. I don't mean to claim that
these results have no value, but I do strongly encourage the authors to
communicate more strongly about how these results should influence
others downstream.

**Broader Impact Concerns:**

No broader impact concerns.

**Claims And Evidence:**

Yes

**Claims Explanation:**

The claims of this paper are all theoretical, described by precise
claims and supported by correct proofs.

**Requested Changes:**

1.  Nit: the paper uses a plethora of acronyms that are unfamiliar (to
    me, at least), which makes reading a little more difficult; for
    instance, BAM, SMP, MSA, etc. I regularly have to return to earlier
    sections to remind myself what these acronyms mean. I think it would
    be best to spell out some of these terms, particularly those that
    don't appear repetitively throughout all sections of the text (BAM,
    MSA, SMP).
2.  Section 4 seems misplaced. From what I can tell, none of the
    contents of this section are contributions of this work, this is all
    background material. This actually makes the text harder to follow,
    because it is not as clear which parts are directly pertinent to the
    goals of the paper.
3.  In the abstract, it says "These results are also of independent
    interest to the stochastic control community, providing new
    implementable objectives \[…\] for SMP-based iterations in
    stochastic control problems". This sounds important, but it was
    barely addressed in the paper. It would really help to see a
    concrete example of this.
4.  My understanding of section 3 is that your derivation of Theorem 1 /
    Theorem 2 suggests that one doesn't need to restrict to
    control-affine dynamics. If so, I would like to see more discussion
    about this as well. What is there to be gained by generalizing the
    class of dynamics (e.g., usually in diffusion models we have total
    freedom over what these dynamics are, why should we make them more
    complicated?). In control theory, there are generally some very nice
    properties of control-affine dynamics w.r.t. deriving closed-form
    solutions to certain equations. Is the same true in this setting? In
    other words, are there also *challenges* involved with generalizing
    the class of dynamics algorithmically?

---

### Review · Reviewer_NHou · 2026-05-22

**Summary Of Contributions:**

This paper revisits adjoint matching from the perspective of stochastic optimal control (SOC). Specifically, it aims to help the community understand adjoint sampling from a theoretically grounded perspective via the stochastic maximum principle (SMP). Motivating applications include sampling from tilted distributions and reward-based fine-tuning of flow/diffusion models, both of which can be casted as SOC problems under SDE constraints. The main contributions of this paper can be summarized from the following aspects:

(1) The paper introduces a basic adjoint matching (BAM) loss for general SOC problems with control-dependent drift and diffusion and convex running costs. The key claim is that the expected BAM objective has the same first variation as the original SOC objective, so its critical points should satisfy the corresponding stationarity condition. This formulation uses both first- and second-order adjoint processes as unbiased estimators of the gradient and Hessian of the cost-to-go.

(2) For the special case when the diffusion term depends on time only, the BAM framework introduced in this paper then derives a simpler lean adjoint matching (AM) loss that avoids second-order terms. The authors then prove that any critical point of the expected lean AM loss satisfies the pointwise stationarity condition for the simplified Hamiltonian, which recovers the optimal control under assumptions of convexity and uniqueness. The original adjoint matching loss can also be shown to arise as a special case of this generic theorem.

(3) Beyond characterizing the critical points of the loss, the paper studies whether gradient-based optimization of the lean AM objective corresponds to a principled improvement procedure. Specifically, this paper shows that gradient flow of the expected lean AM loss coincides with a continuous-time method of successive approximations (MSA) induced by the SMP for SOC, which also doesn't need explicit access to the martingale integrand in the SMP adjoint BSDE. This gives a conceptual explanation for why adjoint matching yields an implementable optimization rule compared to direct SMP-based algorithms, which are generally impractical in high dimensions.

**Additional Comments:**

NA

**Audience:**

Yes

**Audience Explanation:**

This paper studies the intersection of several active topics in TMLR: diffusion and flow-based sampling or generative modeling, post-training or reward-based fine-tuning, and stochastic optimal control. To the best of the reviewer's knowledge, this paper is one of the first studies that provides a clear and theoretically grounded way to explain why the loss function used in adjoint matching works well in practice, so the reviewer expects that TMLR readers will be interested in the paper’s results.

**Claims And Evidence:**

Yes

**Claims Explanation:**

This paper is a theoretical paper with rigorous mathematical arguments and proofs. For instance, Theorem 2 gives a clean specialization to the practically important case with detailed proofs and derivations. Theorem 4, on the other hand, provides a principled interpretation of the gradient flow of lean AM as a continuous-time MSA update, which is the key conceptual bridge claimed by the paper. Overall, the reviewer thinks that all the mathematical and theoretical arguments in this paper are well-supported.

**Requested Changes:**

The reviewer is definitely happy to recommend acceptance of this paper, as it provides a mathematically interesting and elegant explanation for the loss function adopted in adjoint matching. However, this paper can be potentially improved from the following aspects:

(1) Would it be possible for the authors to add a mini-experiment to justify the theoretical findings? The reviewer fully agrees that the BAM objective is mathematically appealing, but it would be great if the authors could further clarify how it can be used in practice, especially for the cases when second-order information of the matrix SDE is needed (Even an experiment on the Gaussian mixture case would be great).

(2) One other possible modification that the authors may consider is to further clarify the assumptions required for optimality claims.
In particular, the reviewer thinks that thsse assumptions are fairly strong in settings like nonlinear or neural-network parameterized models. Would it be possible for the authors to comment on these cases?

(3) Moreover, it seems to the reviewer that this manuscript could be further enhanced by discussing a few more related and future work. For instance, a few variants of adjoint sampling based on SOC or Schrödinger Bridge formulation have been recently developed to tackle sampling problems, such as [1,2,3,4,5,6]. Would it be possible for the authors to comment on how this work could be potentially extended to other settings like SOC of discrete data? Overall, the authors may refer to [2,3,7] for a more comprehensive review of related work (especially work on learning-based sampling methods).

References:

[1] Liu, G. H., Choi, J., Chen, Y., Miller, B. K., & Chen, R. T. (2026). Adjoint schrödinger bridge sampler. Advances in Neural Information Processing Systems, 38, 15673-15708.

[2] Zhu, Y., Guo, W., Choi, J., Liu, G. H., Chen, Y., & Tao, M. (2026). Mdns: Masked diffusion neural sampler via stochastic optimal control. Advances in Neural Information Processing Systems, 38, 35260-35308.

[3] Guo, W., Zhu, Y., Du, X., Nam, J., Chen, Y., Gómez-Bombarelli, R., ... & Choi, J. (2026). Discrete Adjoint Schr\" odinger Bridge Sampler. arXiv preprint arXiv:2602.08243.

[4] So, O., Karrer, B., Fan, C., Chen, R. T., & Liu, G. H. (2026). Discrete adjoint matching. arXiv preprint arXiv:2602.07132.

[5] Park, B., Lee, J., & Liu, G. H. (2025). Functional Adjoint Sampler: Scalable Sampling on Infinite Dimensional Spaces. arXiv preprint arXiv:2511.06239.

[6] Domingo-Enrich, C., Du, Y., & Albergo, M. S. (2026). A unified perspective on fine-tuning and sampling with diffusion and flow models. arXiv preprint arXiv:2605.00229.

[7] Ren, Y., Gao, W., Ying, L., Rotskoff, G. M., & Han, J. (2025). Driftlite: Lightweight drift control for inference-time scaling of diffusion models. arXiv preprint arXiv:2509.21655.

---

### Review · Reviewer_R626 · 2026-05-25

**Summary Of Contributions:**

The paper considers the problem of fine-tuning generative diffusion and flow models to maximize rewards or sample from unnormalized target densities and study it as a high-dimensional Stochastic Optimal Control (SOC) problem. The paper talks about applying the Stochastic Maximum Principle (SMP) to this setting and the challenge of intractable Backward Stochastic Differential Equations (BSDEs) which makes it hard to use iterative trajectory-based solvers like the Method of Successive Approximations (MSA). The paper proposes a Hamiltonian-based theoretical generalization of a recently introduced method called "Adjoint Matching" (AM). For generative models with purely time-dependent noise, the paper proves that this BAM objective reduces to a "Lean AM" loss where second-order terms vanish. Finally, the paper demonstrates analytically that gradient descent on the expected Lean AM loss coincides precisely with a continuous-time MSA update that structurally bypasses the intractable BSDE martingale term. The evaluation is purely theoretical.

**Audience:**

No

**Audience Explanation:**

My main concern with the paper is whether it should be submitted to better suited for a control theory venue. It will be hard to understand the relevance of this work and appreciate the advances by regular machine learning community because of the lots of background tools from control theory required. This is especially important because the paper essentially derives adjoint matching approach as a special case.  It would be good to provide ahistorical overview of prior work in the area and the key challenges being addressed here and how they might help in the future of diffusion modeling or flow matching

**Claims And Evidence:**

Yes

**Claims Explanation:**

- The paper has a good aim of trying to making adjoint matching more rigorous.

- Key theorems are sound and conceptually convincing.

**Requested Changes:**

Please see relevance question above.

---

### Decision · Action_Editor_L69J · 2026-07-09

**Recommendation:** Accept as is

**Audience:**

Yes

**Audience Explanation:**

The reviewers considered the paper to be a bit borderline in terms of audience, with writing style more suitable for control theory venues. However, the paper is still clearly motivated by key machine learning methods (flow matching, diffusion models, sampling from Boltzmann distributions) and the revision version makes the connections to ML literature more transparent, and the numerical experiments are done with models relevant for the ML audience. The paper may remain somewhat difficult to read for broader ML audience, but it certainly is interesting for some readers in the community.

**Claims And Evidence:**

Yes

**Claims Explanation:**

The paper is largely theoretical, explaining how reward tuning and Boltzmann sampling, as used in flow matching and diffusion models, can be re-interpreted as stochastic optimal control. The main contributions are formal theorems that are stated clearly and complemented with formal proofs, and the reviewers did not see weaknesses in the technical development. The key ideas are illustrated using a range of simple numerical examples.